# One-Step Diffusion Distillation through Score Implicit Matching

**Weijian Luo**[*]
Peking University
luoweijian@stu.pku.edu.cn

**Zemin Huang**
Westlake University
huangzemin@westlake.edu.cn

**Zhengyang Geng**
Carnegie Mellon University
zgeng2@cs.cmu.edu

**J. Zico Kolter**
Carnegie Mellon University
zkolter@cs.cmu.edu

**Guo-jun Qi**[†]
Westlake University
guojunq@gmail.com

https://github.com/maple-research-lab/SIM

## Abstract

Despite their strong performances on many generative tasks, diffusion models require a large number of sampling steps in order to generate realistic samples. This has motivated the community to develop effective methods to distill pre-trained diffusion models into more efficient models, but these methods still typically require few-step inference or perform substantially worse than the underlying model. In this paper, we present Score Implicit Matching (SIM) a new approach to distilling pre-trained diffusion models into single-step generator models, while maintaining almost the same sample generation ability as the original model as well as being data-free with no need of training samples for distillation. The method rests upon the fact that, although the traditional score-based loss is intractable to minimize for generator models, under certain conditions we *can* efficiently compute the *gradients* for a wide class of score-based divergences between a diffusion model and a generator. SIM shows strong empirical performances for one-step generators: on the CIFAR10 dataset, it achieves an FID of 2.06 for unconditional generation and 1.96 for class-conditional generation. Moreover, by applying SIM to a leading transformer-based diffusion model, we distill a single-step generator for text-to-image (T2I) generation that attains an aesthetic score of 6.42 with no performance decline over the original multi-step counterpart, clearly outperforming the other one-step generators including SDXL-TURBO of 5.33, SDXL-LIGHTNING of 5.34 and HYPER-SDXL of 5.85. We will release this industry-ready one-step transformer-based T2I generator along with this paper.

## 1 Introduction

Over the past years, diffusion models (DMs) [21, 67, 65] have shown significant advancements across a broad spectrum of applications, ranging from data synthesis [25, 26, 51, 52, 22, 56, 23, 31], to density estimation [32, 8], text-to-image generation[54, 60, 2, 80, 7], text-to-3D creation [56, 74, 28, 34], image editing [47, 9, 19, 1, 30, 49], and beyond [83, 79, 5, 85, 18, 59, 14, 73, 89, 72, 42, 78, 44, 84, 13, 11, 46, 16, 71, 55, 10]. From a high level point of view, diffusion models, also framed as score-based diffusion models, use diffusion processes to corrupt the data distribution. They are then trained to approximate the score functions of the noisy data distributions across varying noise levels.

---

[*]Alternative email: pkulwj1994@icloud.com.

[†]Correspondence to Guo-jun Qi. The project was initiated and supported by the MAPLE lab of Westlake University.

38th Conference on Neural Information Processing Systems (NeurIPS 2024).

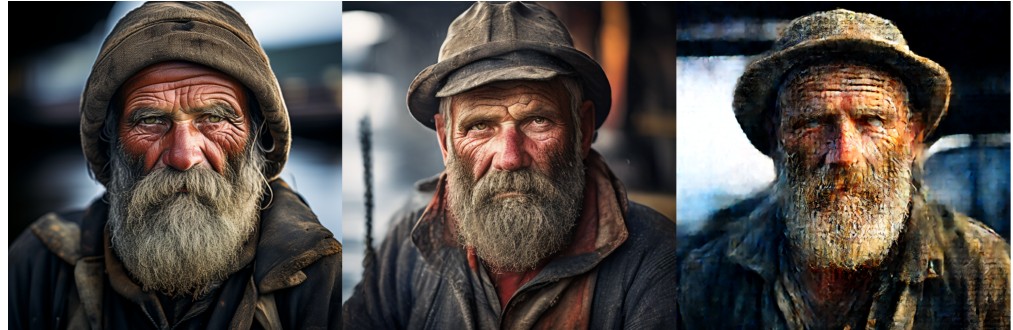

"Seasoned fisherman portrait, weathered skin etched with deep wrinkles, white beard, piercing gaze beneath a fisherman's hat, softly blurred dock background accentuating rugged features, captured under natural light, ultra-realistic, high dynamic range photo."

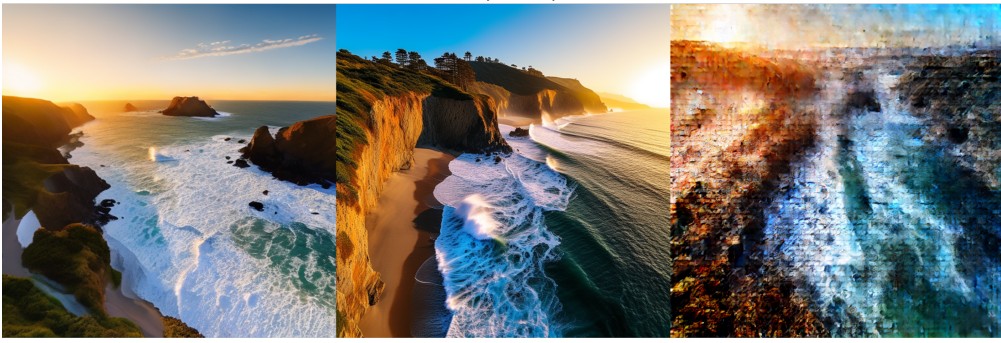

"Drone view of waves crashing against the rugged cliffs along Big Sur's garay point beach. The crashing blue waters create white-tipped waves, while the golden light of the setting sun illuminates the rocky shore. A small island with a lighthouse sits in the distance, and green shrubbery covers the cliff's edge. The steep drop from the road down to the beach is a dramatic feat, with the cliff's edges jutting out over the sea. This is a view that captures the raw beauty of the coast and the rugged landscape of the Pacific Coast Highway."

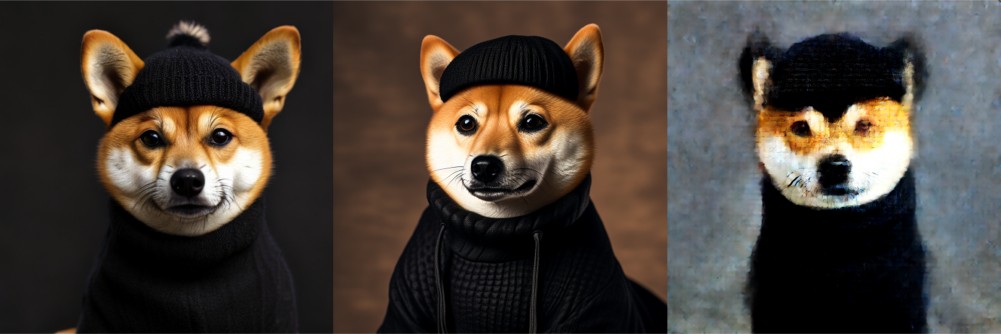

"A Shiba Inu dog wearing a beret and black turtleneck."

Figure 1: Time for a Human Preference Study! Could you please tell us which one is better? Hint: the rightmost column is the one-step Latent Consistency Model of PixelArt-$\alpha$; The left two columns are randomly placed, with one generated from our one-step **SIM-DiT-600M** model, and another generated from the 14-step PixelArt-$\alpha$ teacher diffusion model. We put the answer in Appendix B.1.

Diffusion models have multiple advantages, such as training flexibility, scalability, and the ability to produce high-quality samples, making them a favored choice for modern AIGC models. After training, the learned score functions can be used to reverse the data corruption process, which can be implemented by numerically solving the associated stochastic differential equation. Such a data generation mechanism usually requires many neural network evaluations, which leads to a significant limitation of DMs: *the generation performance of DMs degrades substantially when the number of sampling steps is reduced*. This shortcoming restricts the practical deployment of DMs, particularly where quick inference is crucial, such as on devices with limited computational capacities like mobile phones and edge devices, or in applications requiring rapid response times.

This challenge has spurred a variety of approaches aimed at expediting the sampling process of diffusion models while preserving their robust generative capabilities. Distillation approaches, in particular, focus on applying distillation algorithms to transition the knowledge from pre-trained, teacher diffusion models to efficient student-generative models which are capable of producing high-quality samples within a few generation steps.

Some works have studied the diffusion distillation algorithm through the lens of probability divergence minimization. For instance, Luo et al. [43], Yin et al. [82] have studied the algorithms that minimize the KL divergence between teacher and one-step student models. Zhou et al. [93] have explored distilling with Fisher divergences, resulting in impressive empirical performances. Though these studies have contributed to the community in both theoretical and empirical aspects with applicable single-step generator models, their theories are built upon specific divergences, namely the Kullback-Leibler divergence and the Fisher divergence, which potentially restrict the distillation performances. A more general framework for understanding and improving diffusion distillation is still lacking.

In this work, we introduce Score Implicit Matching (SIM), a novel framework for distilling pre-trained diffusion models into one-step generator networks while maintaining high-quality generations. To do so, we propose a wide and flexible class of score-based divergences between the (intractable) score function of the generator model and that of the original diffusion model, for arbitrary distance functions between the two score functions. The key technical insight of this work is that although such divergences cannot be computed explicitly, the *gradient* of these divergences *can* be computed exactly using a result we call the *score-gradient theorem*, leading to an implicit minimization of the divergence. This lets us efficiently train models based on such divergences.

We evaluate the performance of SIM compared to previous approaches, using different choices of distance functions to define the divergence. Most relatedly, we compare SIM with the Diff-Instruct (DI) [43] method, which uses a KL-based divergence term, and the Score Identity Distillation (SiD) method [93], which we show to be a special case of our approach when the distance function is simply chosen to be the squared $L_2$ distance (though derived in an entirely different fashion). We also show empirically that SIM with a specially-designed Pseudo-Huber distance function shows faster convergences and stronger robustness to hyper-parameters than $L_2$ distance, making the resulting method substantially strong than previous approaches.

Finally, we show that SIM obtains very strong empirical performance in absolute terms relative to past work in the field on CIFAR10 image generation and text-to-image generation. On the CIFAR10 dataset, SIM shows a one-step generative performance with a Frechet Inception Distance (FID) of 2.06 for unconditional generation and 1.96 for class-conditional generation. More qualitatively, distilling a leading diffusion-transformer-based [53] text-to-image diffusion model results in an extremely capable one-step text-to-image generator which we show is almost lossless in terms of generative performances as teacher diffusion model. Particularly, by applying SIM to PixelArt-$\alpha$ [7], a single-step generator is distilled that reaches an outstanding aesthetic score of $6.42$ with no performance decline over the original multi-step diffusion model. This remarkably outperforms the other one-step text-to-image generators including SDXL-TURBO [64] of 5.33, SDXL-LIGHTNING [35] of 5.34 and HYPER-SDXL [57] of 5.85. Such a result not only marks a new direction for one-step text-to-image generation but also motivates further studies of distilling diffusion-transformer-based AIGC models in other domains such as video generation.

## 2 Diffusion Models

In this section, we introduce preliminary knowledge and notations about diffusion models and diffusion distillation. Assume we observe data from the underlying distribution $q_d(\boldsymbol{x})$. The goal of generative modeling is to train models to generate new samples $\boldsymbol{x} \sim q_d(\boldsymbol{x})$. The forward diffusion process of DM transforms any initial distribution $q_0 = q_d$ towards some simple noise distribution,

$$\mathrm{d}\boldsymbol{x}_t = \boldsymbol{F}(\boldsymbol{x}_t, t)\mathrm{d}t + G(t)\mathrm{d}\boldsymbol{w}_t, \qquad (2.1)$$

where $\boldsymbol{F}$ is a pre-defined drift function, $G(t)$ is a pre-defined scalar-value diffusion coefficient, and $\boldsymbol{w}_t$ denotes an independent Wiener process. A continuous-indexed score network $\boldsymbol{s}_\varphi(\boldsymbol{x}, t)$ is employed to approximate marginal score functions of the forward diffusion process (2.1). The learning of score networks is achieved by minimizing a weighted denoising score matching objective [70, 67],

$$\mathcal{L}_{DSM}(\varphi) = \int_{t=0}^{T} \lambda(t)\mathbb{E}_{\boldsymbol{x}_0 \sim q_0, \boldsymbol{x}_t|\boldsymbol{x}_0 \sim q_t(\boldsymbol{x}_t|\boldsymbol{x}_0)} \|\boldsymbol{s}_\varphi(\boldsymbol{x}_t, t) - \nabla_{\boldsymbol{x}_t} \log q_t(\boldsymbol{x}_t|\boldsymbol{x}_0)\|_2^2 \mathrm{d}t. \qquad (2.2)$$

Here the weighting function $\lambda(t)$ controls the importance of the learning at different time levels and $q_t(\boldsymbol{x}_t|\boldsymbol{x}_0)$ denotes the conditional transition of the forward diffusion (2.1). After training, the score network $\boldsymbol{s}_\varphi(\boldsymbol{x}_t, t) \approx \nabla_{\boldsymbol{x}_t} \log q_t(\boldsymbol{x}_t)$ is a good approximation of the marginal score function of the diffused data distribution. High-quality samples from a DM can be drawn by simulating SDE which

is implemented by the learned score network [67]. However, the simulation of an SDE is significantly slower than that of other models such as one-step generator models.

# 3 Score Implicit Matching

In this section, we introduce Score Implicit Matching which is a general method tailored for the one-step distillation of score-based diffusion models. We first introduce the problem setup and notations, then introduce a general family of score-based probability divergences and show how SIM can be used to minimize the mentioned divergences. We finally discuss specific choices of the method, such as the choice of distance function, and explore the effect this has on the distillation.

**Problem setup.** Our starting point is a pre-trained diffusion model specified by the score function

$$
s_{q_t}(\boldsymbol{x}_t) := \nabla_{\boldsymbol{x}_t} \log q_t(\boldsymbol{x}_t) \tag{3.1}
$$

where $q_t(\boldsymbol{x}_t)$'s are the underlying distribution diffused at time $t$ according to (2.1). We assume that the pre-trained diffusion model provides a sufficiently good approximation of data distribution, and thus will be the only item of consideration for our approach.

The student model of interest is a single-step generator network $g_\theta$, which can transform an initial random noise $\boldsymbol{z} \sim p_z$ to obtain a sample $\boldsymbol{x} = g_\theta(\boldsymbol{z})$; this network is parameterized by network parameters $\theta$. Let $p_{\theta,0}$ denote the data distribution of the student model, and $p_{\theta,t}$ denote the marginal diffused data distribution of the student model with the same diffusion process (2.1). The student distribution implicitly induces a score function

$$
s_{p_{\theta,t}}(\boldsymbol{x}_t) := \nabla_{\boldsymbol{x}_t} \log p_{\theta,t}(\boldsymbol{x}_t), \tag{3.2}
$$

and evaluating it is generally performed by training an alternative score network as elaborated later.

## 3.1 General Score-based Divergences

The goal of one-step diffusion distillation is to let the student distribution $p_{\theta,0}$ match the data distribution $q_0$. To do so, we propose to match the diffused marginal distribution $p_{\theta,t}$ and $q_t$ at all diffusion time levels. We can define such an objective via the following general score-based divergence. Assume $\mathbf{d} : \mathbb{R}^d \to \mathbb{R}$ is a scalar-valued proper distance function (i.e., a function that obeys $\forall \boldsymbol{x}, \mathbf{d}(\boldsymbol{x}) \geq 0$ and $\mathbf{d}(\boldsymbol{x}) = 0$ if and only if $\boldsymbol{x} = \boldsymbol{0}$). Given a sampling distribution $\pi_t$ that has larger distribution support than $p_t$ and $q_t$, we can formally define a time-integral score divergence as

$$
\mathcal{D}^{[0,T]}(p, q) := \int_{t=0}^{T} w(t) \mathbb{E}_{\boldsymbol{x}_t \sim \pi_t} \left\{ \mathbf{d}(s_{p_t}(\boldsymbol{x}_t) - s_{q_t}(\boldsymbol{x}_t)) \right\} \mathrm{d}t, \tag{3.3}
$$

where $p_t$ and $q_t$ denote the marginal densities of the diffusion process (2.1) at time $t$ initialized with $q$ and $p$ respectively. $w(t)$ is an integral weighting function. Clearly, we have $\mathcal{D}^{[0,T]}(p, q) = 0$ if and only if all marginal score functions agree, which implies that $p_0(\boldsymbol{x}_t) = q_0(\boldsymbol{x}_t)$, $a.s.$ $\pi_0$.

## 3.2 Score Implicit Matching

Based upon this motivation, we would like to minimize the integral score-based divergence between $p_\theta$ and $q$ in order to train the student model, i.e.,

$$
\mathcal{L}(\theta) = \mathcal{D}^{[0,T]}(p_\theta, q) = \int_{t=0}^{T} w(t) \mathbb{E}_{\boldsymbol{x}_t \sim \pi_t} \left[ \mathbf{d}(s_{p_{\theta,t}}(\boldsymbol{x}_t) - s_{q_t}(\boldsymbol{x}_t)) \right] \mathrm{d}t, \tag{3.4}
$$

where we assume that the distribution $\pi_t$ has no parameter dependence of $\theta$, such as $\psi_t(\boldsymbol{x}_t) = p_{\mathrm{sg}[\theta]}(\boldsymbol{x}_t)$. Taking the gradient with respect to $\theta$, we have

$$
\frac{\partial}{\partial \theta} \mathcal{L}(\theta) = \int_{t=0}^{T} w(t) \mathbb{E}_{\boldsymbol{x}_t \sim \pi_t} \left[ \mathbf{d}'(s_{p_{\theta,t}}(\boldsymbol{x}_t) - s_{q_t}(\boldsymbol{x}_t)) \frac{\partial}{\partial \theta} s_{p_{\theta,t}(\boldsymbol{x}_t)} \right] \mathrm{d}t, \tag{3.5}
$$

where $\mathbf{d}'$ denotes the derivative of $\mathbf{d}$ wrt. its inputs, i.e. $\nabla_{\boldsymbol{y}} \mathbf{d}(\boldsymbol{y})$. Unfortunately, because the score function is not tractable, it is impossible to compute $\frac{\partial}{\partial \theta} s_{p_{\theta,t}(\boldsymbol{x}_t)}$ directly, rendering such a direct approach impractical.

---

**Algorithm 1:** Score Implicit Matching for Diffusion Distillation. (Pseudo-code in Appendix A.2)

---

**Input:** pre-trained DM $s_{q_t}(.)$, generator $g_\theta$, prior distribution $p_z$, online DM $s_\psi(.)$;
     differentiable distance function $\mathbf{d}(.)$, and forward diffusion (2.1).

**while** *not converge* **do**

   with frozen $\theta$, update $\psi$ using SGD with gradient

$$\mathrm{Grad}(\psi) = \frac{\partial}{\partial \psi} \int_{t=0}^{T} \lambda(t) \mathbb{E}_{\substack{z \sim p_z, x_0 = g_\theta(z), \\ x_t | x_0 \sim q_t(x_t | x_0)}} \| s_\psi(x_t, t) - \nabla_{x_t} \log q_t(x_t | x_0) \|_2^2 \mathrm{d}t.$$

   with frozen $\psi$, update $\theta$ using SGD with the gradient

$$\mathrm{Grad}(\theta) = \frac{\partial}{\partial \theta} \int_{t=0}^{T} w(t) \mathbb{E}_{\substack{z \sim p_z, x_0 = g_\theta(z), \\ x_t | x_0 \sim q_t(x_t | x_0)}} \left\{ -\mathbf{d}'(y_t) \right\}^T \left\{ s_\psi(x_t, t) - \nabla_{x_t} \log q_t(x_t | x_0) \right\} \mathrm{d}t,$$

   where $y_t := s_\psi(x_t, t) - s_{q_t}(x_t)$.

**end**

**return** $\theta, \psi$.

---

Fortunately, a key finding of our paper is if we choose the sampling distribution to the diffused implicit distribution, i.e. $\pi_t = p_{\mathrm{sg}[\theta],t}$ where the notation $\mathrm{sg}[\theta]$ denotes the *stop gradient* operator that cuts off the parameter dependence of $\theta$, the loss function (3.4) along with its intractable gradient (3.5) can be minimized efficiently via an gradient-equivalent loss. This relies on our Theorem 3.1.

**Theorem 3.1** (Score-divergence gradient Theorem). If distribution $p_{\theta,t}$ satisfies some mild regularity conditions, we have for any score function $s_{q_t}(.)$, the following equation holds for all parameter $\theta$:

$$\mathbb{E}_{x_t \sim p_{\mathrm{sg}[\theta],t}} \left[ \mathbf{d}'(s_{p_{\theta,t}}(x_t) - s_{q_t}(x_t)) \frac{\partial}{\partial \theta} s_{p_{\theta,t}}(x_t) \right] \tag{3.6}$$

$$= -\frac{\partial}{\partial \theta} \mathbb{E}_{\substack{x_0 \sim p_{\theta,0}, \\ x_t | x_0 \sim q_t(x_t | x_0)}} \left[ \left\{ \mathbf{d}'(s_{p_{\mathrm{sg}[\theta],t}}(x_t) - s_{q_t}(x_t)) \right\}^T \left\{ s_{p_{\mathrm{sg}[\theta],t}}(x_t) - \nabla_{x_t} \log q_t(x_t | x_0) \right\} \right].$$

The key observation here is that we replace the intractable *gradient* of the score function on the left-hand side of (3.6) with a much affordable *evaluation* of the score function on the right-hand side, the latter of which can be accomplished much more easily using a separate approximation network. This theorem can be proved by using score-projection identity [70, 93] which was first introduced to bridge denoising score matching with denoising auto-encoders. However, the key in proving Theorem 3.1 is a proper choice of $\theta$-parameter (in)dependence by appropriately stopping the gradients shown in this theorem. We provide the detailed proof in Appendix A.1.

Now it is ready to reveal the objective we will use to train the implicit generator $g_\theta$. A direct result of (3.6) is the gradient (3.5) can be realized via minimizing a tractable loss function

$$\mathcal{L}_{SIM}(\theta) = \int_{t=0}^{T} w(t) \mathbb{E}_{\substack{z \sim p_z, x_0 = g_\theta(z), \\ x_t | x_0 \sim q_t(x_t | x_0)}} \left\{ -\mathbf{d}'(y_t) \right\}^T \left\{ s_{p_{\mathrm{sg}[\theta],t}}(x_t) - \nabla_{x_t} \log q_t(x_t | x_0) \right\} \mathrm{d}t \tag{3.7}$$

with $y_t := s_{p_{\mathrm{sg}[\theta],t}}(x_t) - s_{q_t}(x_t)$. By Theorem 3.1, this alternative loss has an identical gradient to that of the original loss without the need to access the gradient of the score network.

In practice, we can use another online diffusion model $s_\psi(x_t, t)$ to approximate the generator model's score function $s_{p_{\mathrm{sg}[\theta],t}}(x_t)$ pointwise, which was also done in previous works such as Luo et al. [43], Zhou et al. [93], and Yin et al. [82]. *We name the distillation method that minimizes the objective $\mathcal{L}_{SIM}(\theta)$ in (3.7) the Score Implicit Matching (SIM) because the learning process implicitly matches the intractable marginal score function $s_{p_{\theta,t}}(.)$ of the implicit student model with the explicit score function of the pre-trained diffusion model $s_{q_t}(.)$.*

The complete algorithm for SIM is shown in Algorithm 1, which trains the student model through two alternative phases between learning the marginal score function $s_\psi$, and updating the generator model with gradient (3.7). The former phase follows the standard DM learning procedure, i.e., minimizing the denoising score matching loss function (2.2), with a slight change that the sample is generated from the generator. The resulting $s_\psi(x_t, t)$ provides a good pointwise estimation of $s_{p_{\mathrm{sg}[\theta],t}}(x_t)$.

The latter phase updates the generator's parameter $\theta$ by minimizing the loss function (3.7), where two needed functions are provided by pretrained DM $\boldsymbol{s}_{q_t}(\boldsymbol{x}_t)$ and learned DM $\boldsymbol{s}_\psi(\boldsymbol{x}_t, t)$.

### 3.3  Instances of Score Implicit Matching.

The previous section introduced the SIM algorithm without choosing a specific distance function $\mathbf{d}(.)$. Here we discuss different choices and their influence on the distillation process. We also show that in the SIM framework, the SiD can be viewed as a special case.

**The Design Choice of Distance Function** $\mathbf{d}(.)$.  Clearly, various choices of distance function $\mathbf{d}(.)$ result in different distillation algorithms. Perhaps the most natural choice of the distance function is a simple squared distance, i.e. $\mathbf{d}(\boldsymbol{y}_t) = \|\boldsymbol{y}_t\|_2^2$. The corresponding derivative term writes $\mathbf{d}'(\boldsymbol{y}_t) = 2\boldsymbol{y}_t$. In fact, such a loss function recovers the *delta loss* studied in SiD [93], in which the authors empirically find that such a loss function works satisfactorily (though through a very different derivation). Thus, SiD is in fact a special case of SIM, though the derivation of SiD there does not suggest how alternative losses may be employed. A direct generalization of the quadratic form is the $\alpha$-power of the $\alpha$-norm where $\alpha > 1$ and $\alpha$ is even. In this case, the distance function writes $\mathbf{d}(\boldsymbol{y}_t) = \alpha \boldsymbol{y}_t^{(\alpha-1)}$ and the resulting loss function is summarized in Table 4 in Appendix A.3.

**The Pseudo-Huber distance function.**  Different from powered norms, we introduce SIM with the Pseudo-Huber distance function, which is defined with $\boldsymbol{d}(\boldsymbol{y}) := \sqrt{\|\boldsymbol{y}_t\|_2^2 + c^2} - c$, where $c$ is a pre-defined positive constant. The corresponding distillation objective writes

$$\mathcal{L}_{SIM}(\theta) = -\left\{\frac{\boldsymbol{y}_t}{\sqrt{\|\boldsymbol{y}_t\|_2^2 + c^2}}\right\}^T \left\{\boldsymbol{s}_\psi(\boldsymbol{x}_t, t) - \nabla_{\boldsymbol{x}_t} \log q_t(\boldsymbol{x}_t|\boldsymbol{x}_0)\right\}. \tag{3.8}$$

*In the rest of this paper, we will use the Pseudo-Huber distance as the default choice of the distance, unless specified otherwise.* Due to the limited space, we summarize different choices of distance function and the corresponding loss functions in Table 4 as well as their derivations, along with more discussions in Appendix A.3.

Particularly, unlike SiD (the $L^2$ case in Table 4), with the Pseudo-Huber distance in the SIM, we observe that the vector $\boldsymbol{y}_t$ is naturally normalized adaptively by dividing by a squared root of the vector. Such a normalization can stabilize the training loss, resulting in a robust and fast-converging distillation process. In section 4.1, we conduct empirical experiments to show three advantages: robustness to large-learning rate, fast convergence, and improved performances.

### 3.4  Related Works

Diffusion distillation [41] is a research area that aims to reduce generation costs using teacher diffusion models. It involves three primary distillation methods: 1) *Trajectory Distillation:* This method trains a student model to mimic the generation process of diffusion models with fewer steps. Direct distillation ([39, 15]) and progressive distillation ([61, 48]) variants predict less noisy data from noisy inputs. Consistency-based methods ([68, 29, 66, 36, 17]) minimize the self-consistency metric. These require true data samples for training. 2) *Distributional Matching:* It focuses on aligning the student's generation distribution with that of a teacher diffusion model. Among them are adversarial training methods ([76, 77]) requiring real data for distilling diffusion models. Another important line of methods attempts to minimize divergences like KL ([82]) such as Diff-Instruct (DI) [45, 82] and Fisher divergence such as Score identity Distillation (SiD) ([93]), often without needing real data. Though SIM has gotten inspiration from SiD and DI, the gap between SIM and SiD and DI is significant. SIM not only offers solid mathematical foundations which may lead to a deep understanding of diffusion distillation, but also provides substantial flexibility in using different distance functions, resulting in strong empirical performances when using specific Pseudo-Huber distance. 3) *Other Methods:* Methods like operator learning ([86]), ReFlow ([37]), and FMM [3] provide alternative insights into distillation. Moreover, many works made outstanding efforts to scale up diffusion distillation to one-step text-to-image generation and beyond[40, 50, 69, 82, 92]

Table 1: Unconditional sample quality on CIFAR-10. † means method we reproduced.

| METHOD | NFE ($\downarrow$) | FID ($\downarrow$) |
|---|---|---|
| **DIFFERENT ARCHITECTURE AS EDM MODEL** | | |
| DDPM [21] | 1000 | 3.17 |
| DD-GAN(T=2) [76] | 2 | 4.08 |
| KD [39] | 1 | 9.36 |
| TDPM [90] | 1 | 8.91 |
| DFNO [88] | 1 | 4.12 |
| 3-REFLOW (+DISTILL) [37] | 1 | 5.21 |
| STYLEGAN2-ADA [24] | 1 | 2.92 |
| STYLEGAN2-ADA+DI [43] | 1 | 2.71 |
| **SAME ARCHITECTURE AS EDM[26] MODEL** | | |
| EDM [26] | 35 | 1.97 |
| EDM [26] | 15 | 5.62 |
| PD [61] | 2 | 5.13 |
| CD [68] | 2 | 2.93 |
| GET [15] | 1 | 6.91 |
| CT [68] | 1 | 8.70 |
| ICT-DEEP [66] | 2 | 2.24 |
| DIFF-INSTRUCT [43] | 1 | 4.53 |
| DMD [82] | 1 | 3.77 |
| CTM [29] | 1 | 1.98 |
| CTM[29] | 2 | **1.87** |
| SID ($\alpha = 1.0$) [93] | 1 | 1.92 |
| SID ($\alpha = 1.2$)[93] | 1 | 2.02 |
| **DI**† | 1 | 3.70 |
| **SIM (OURS)** | 1 | 2.06 |

Table 2: Class-conditional sample quality on CIFAR10 dataset. † means method we reproduced.

| METHOD | NFE ($\downarrow$) | FID ($\downarrow$) |
|---|---|---|
| **DIFFERENT ARCHITECTURE AS EDM MODEL** | | |
| BIGGAN [4] | 1 | 14.73 |
| BIGGAN+TUNE[4] | 1 | 8.47 |
| STYLEGAN2 [25] | 1 | 6.96 |
| MULTIHINGE [27] | 1 | 6.40 |
| FQ-GAN [87] | 1 | 5.59 |
| STYLEGAN2-ADA [24] | 1 | 2.42 |
| STYLEGAN2-ADA+DI [43] | 1 | 2.27 |
| STYLEGAN2 + SMART [75] | 1 | 2.06 |
| STYLEGAN-XL [63] | 1 | 1.85 |
| **SAME ARCHITECTURE AS EDM[26] MODEL** | | |
| EDM [26] | 35 | 1.82 |
| EDM [26] | 20 | 2.54 |
| EDM [26] | 10 | 15.56 |
| EDM [26] | 1 | 314.81 |
| GET [15] | 1 | 6.25 |
| DIFF-INSTRUCT [43] | 1 | 4.19 |
| DMD (W.O. REG) [82] | 1 | 5.58 |
| DMD (W.O. KL) [82] | 1 | 3.82 |
| DMD [82] | 1 | 2.66 |
| CTM [29] | 1 | 1.73 |
| CTM[29] | 2 | **1.63** |
| SID ($\alpha = 1.0$) [93] | 1 | 1.93 |
| SID ($\alpha = 1.2$)[93] | 1 | 1.71 |
| **SIM (OURS)** | 1 | 1.96 |

# 4 Experiments

## 4.1 One-step CIFAR10 Generation

**Experiment Settings.** In this experiment, we apply SIM to distill the pre-trained EDM [26] diffusion models into one-step generator models on the CIFAR10 [33] dataset. We follow the same setting as DI [43] and SiD [93] to distill the diffusion model into a one-step generator. Details can be found in Appendix B.2. We refer to the high-quality codebase of SiD [93][3] to reproduce its results by closely referring to its configurations on our devices. We also re-implement the DI under the same experiment settings.

**Performances.** We evaluate the performance of the trained generator via Frechet Inception Distance (FID) [20], which is the lower the better. We refer to the evaluation protocols in [43] for comparison [4]. Table 1 and 2 summarize the FID of generative models on CIFAR10 datasets. We reproduce the SiD and the DI with the same computing environments and evaluation protocol as SIM for a fair comparison. Models in the upper part of the table have different architectures or diffusion models from the EDM model, while the models in the lower part of the tables share exactly the same architecture and the teacher EDM diffusion models, which thus are directly comparable.

As shown in Table 1, for the CIFAR10 unconditional generation task, the proposed SIM achieves a decent FID of 2.06 with only one generation step, outperforming SiD and DI with the same training compute. It is on par with the CTM and the SiD's official implementation which are trained to fully converge with training costs of hundreds of GPU days. For the class-conditional generation in Table 2, the SIM achieves an FID of 1.96, acting among top-performing models.

The CIFAR-10 generation tasks are much toyish as merely performed with diffusion models of limited capacities on a simple dataset. We will perform experiments to distill from top-performing transformer-based diffusion models for text-to-image generation tasks. We will show that the one-step T2I generator distilled by SIM demonstrates state-of-the-art results over other industry-level models.

---

[3] https://github.com/mingyuanzhou/SiD
[4] https://github.com/pkulwj1994/diff_instruct

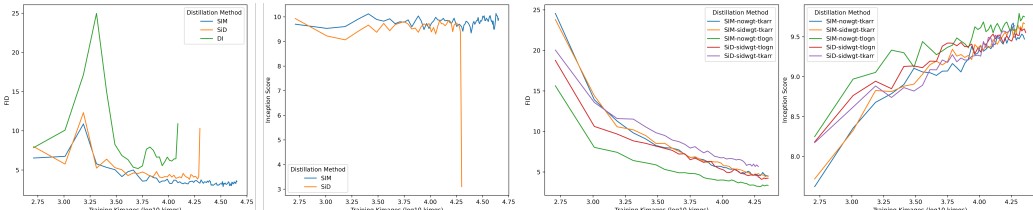

Figure 2: **Left Two:** Comparison of distillation methods with a batch size of 256 and a learning rate of $1e-4$. *(Left):* the FID value. *(Right):* the Inception Scores. **Right Two:** Comparison of distillation methods with a batch size of 256 and a learning rate of $1e-5$. *(Left):* the FID value. *(Right):* the Inception Scores. All methods are constrained to the same settings except for the distillation methods.

Before that let us further look into some advantages of SIM – robustness to large learning rate and faster convergences – over SiD and DI on CIFAR-10, which will shed some light on how distillation methods scale up to more complex tasks with much larger neural networks.

**Robustness to large learning rate.** We apply SIM, SiD, and DI under the same settings to distill from EDM (details in Appendix) on the CIFAR10 unconditional generation task, with a learning rate of $1e-4$, and plot the Fretchet Inception Distance (FID) [20] and the Inception Score [62] in Figure 2. Both the DI and the SiD are unstable even in the early training phase, while the SIM can steadily converge even with a large learning rate. The potential reason is that SIM naturally normalizes the loss objective to keep its scale from changing abruptly along the training process. *This distinguishes SIM from SiD in practice for training large models, because training modern large models is so expensive that researchers often have few chances to adjust the hyperparameters within budget.*

**Fast convergence.** The second advantage of SIM is its faster convergence than SiD [5]. To show this, we follow the same setting as SiD on CIFAR10 unconditional generation. As shown in Figure 2 and Figure **??**, under all configurations, the SIM consistently shows better FID and Inception Scores under the same training iterations. Due to page limitations, we put more details in Appendix B.2.

Experiments on CIFAR10 generation show that SIM is a strong, robust, yet fast converging one-step diffusion distillation algorithm. However, the power of SIM is not restricted to a toy CIFAR-10 benchmark. In section 4.2, we apply the SIM to distill a 0.6B DiT [53]) based text-to-image diffusion model and obtain the state-of-the-art transformer-based one-step generator.

## 4.2 Transformer-based One-step Text-to-Image Generator

**Experiment Settings.** In recent years, transformer-based text-to-X generation models have gained great attention across image generations such as Stable Diffusion V3 [12] and video generation such as Sora [6]. In this section, we apply SIM to distill one of the leading open-sourced DiT-based diffusion models that have gained lots of attention recently: the 0.6B PixelArt-$\alpha$ model [7], which is built upon with DiT model [53], resulting in the state-of-the-art one-step generator in terms of both quantitative evaluation metric and subjective user studies.

**Experiment Settings and Evaluation Metrics.** The goal of one-step distillation is to accelerate the diffusion model into one-generation steps while maintaining or even outperforming the teacher diffusion model's performances. To verify the performance gap between our one-step model and the diffusion model, we compare four quantitative values: the aesthetic score, the PickScore, the Image Reward, and our user-studied comparison score. On the SAM-LLaVA-Caption10M, which is one of the datasets the original PixelArt-$\alpha$ model is trained on, we compare the SIM one-step model, which we called the **SIM-DiT-600M**, with the PixelArt-$\alpha$ model with a 14-step DPM-Solver[38] to evaluate the in-data performance gap. We also compare the SIM-DiT-600M and PixelArt-$\alpha$ with other few-step models, such as LCM [40], TCD [91], PeReflow [81], and Hyper-SD [57] series on the widely used COCO-2017 validation dataset. We refer to Hyper-SD's evaluation protocols to compute evaluation metrics. Table 3 summarizes the evaluation performances of all models. For the human preference study against PixArt-$\alpha$ and SIM-DiT-600M, we randomly select 17 prompts from the SAM Caption dataset and generate images with both PixArt-$\alpha$ and SIM-DiT-600M, then

---

[5]We find that the DI converges fast but suffers from mode-collapse issues. So we do not compare with it.

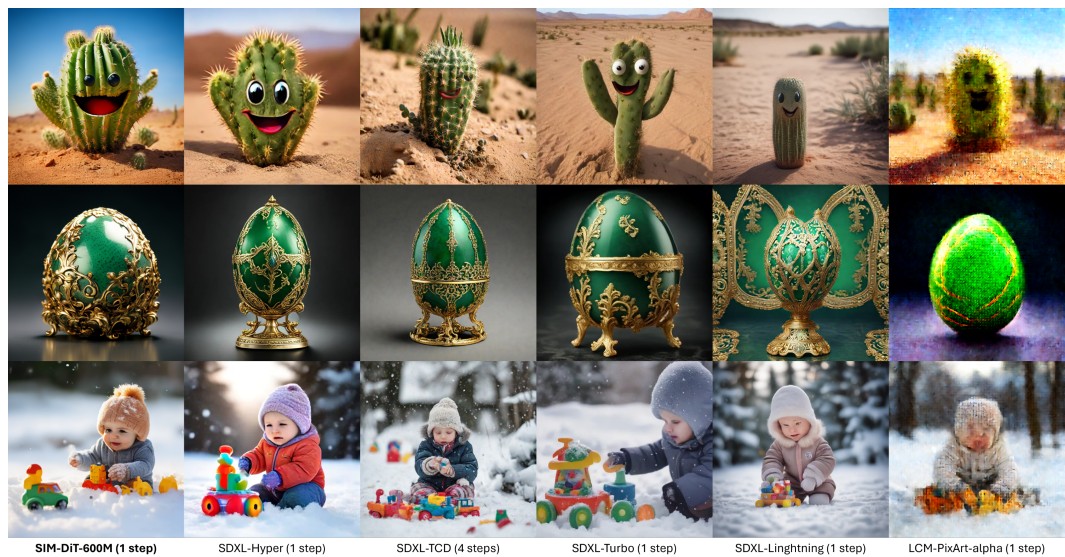

| SIM-DiT-600M (1 step) | SDXL-Hyper (1 step) | SDXL-TCD (4 steps) | SDXL-Turbo (1 step) | SDXL-Linghtning (1 step) | LCM-PixArt-alpha (1 step) |

Figure 3: Qualitative comparison of SIM-DiT-600M against other few-step text-to-image models. Please zoom in to check details, lighting, and aesthetic performances. Prompts in Appendix B.7.

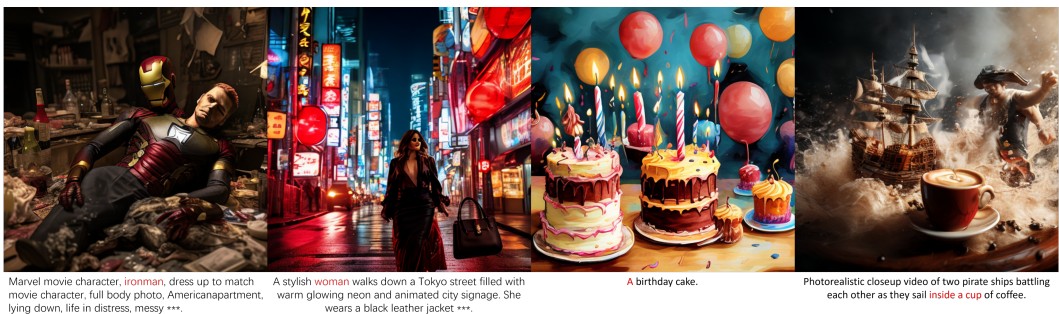

Marvel movie character, ironman, dress up to match movie character, full body photo, Americanapartment, lying down, life in distress, messy ***.

A stylish woman walks down a Tokyo street filled with warm glowing neon and animated city signage. She wears a black leather jacket ***.

A birthday cake.

Photorealistic closeup video of two pirate ships battling each other as they sail inside a cup of coffee.

Figure 4: Visualization of bad generation cases of one-step SIM-DiT model.

ask the studied user to choose their preference according to image quality and alignments with the prompts. Figure 1 shows a visualization of our user study cases, in which it is difficult to distinguish the images from PixArt-$\alpha$ and SIM-DiT-600M.

**Almost lossless one-step distillation.** It is surprising that SIM-DiT-600M achieves almost no performance loss compared to teacher diffusion models. For instance, on the SAM Caption dataset in Table 3, SIM-DiT-600M recovers $99.6\%$ aesthetic score of PixArt-$\alpha$ model and $100\%$ PickScore. However, the SIM-DiT-600M shows a slightly smaller Image Reward, which can be potentially optimized with more training computes. When compared with leading few-step text-to-image models such as SDXL-Turbo, SDXL-lightning, and Hyper-SDXL, the SIM-DiT-600M shows a dominant aesthetic score with a significant margin, together with a decent Image Reward and Pick Score.

Besides the top performance, the training cost of SIM-DiT-600M is surprisingly cheap. Our best model is trained (data-freely) with 4 A100-80G GPUs for 2 days, while other models in Table 3 require hundreds of A100 GPU days. We summarize the distillation costs in Table 3, marking that SIM is a super efficient distillation method with astonishing scaling ability. We believe such efficiency comes from two properties of SIM. First, the SIM is data-free, making the distillation process not need ground truth image data. Second, the use of the Pseudo-Huber distance function (3.3) adaptively normalizes the loss function, resulting in robustness to hyper-parameters and training stability.

**Qualitative comparison.** Figure 3 qualitatively compares SIM-DiT-600M against other leading few-step text-to-image generative models. It is obvious that SIM-DiT-600M generates images with higher aesthetic performances than other models. This reflects the quantitative results in Table 3 where the SIM-DiT-600M reaches a high aesthetic score. Both the quantitative and qualitative results showcase the SIM-DiT-600M as the top-performing one-step text-to-image generator. Please check our supplementary materials for more qualitative evaluations.

| Model | Steps | Type | Params | Aes Score | Image Reward | Pick Score | User Pref | Distill Cost |
|---|---|---|---|---|---|---|---|---|
| SD15-Base [58] | 25 | UNet | 860 M | 5.26 | 0.18 | 0.217 | | |
| SD15-LCM [40] | 4 | UNet | 860 M | 5.66 | -0.37 | 0.212 | | 8 A100× 4 Days |
| SD15-TCD [91] | 4 | UNet | 860 M | 5.45 | -0.15 | 0.214 | | 8 A800× 5.8 Days |
| PeRFlow [81] | 4 | UNet | 860 M | 5.64 | -0.35 | 0.208 | | M GPU× N Days |
| Hyper-SD15[57] | 1 | UNet | 860 M | 5.79 | 0.29 | 0.215 | | 32 A100× N Days |
| SDXL-Base [58] | 25 | UNet | 2.6 B | 5.54 | 0.87 | 0.229 | | |
| SDXL-LCM [40] | 4 | UNet | 2.6 B | 5.42 | 0.48 | 0.224 | | 8 A100× 4 Days |
| SDXL-TCD [91] | 4 | UNet | 2.6 B | 5.42 | 0.67 | 0.226 | | 8 A800× 5.8 Days |
| SDXL-Lightning [35] | 4 | UNet | 2.6 B | 5.63 | 0.72 | 0.229 | | 64 A100× N Days |
| Hyper-SDXL[57] | 4 | UNet | 2.6 B | 5.74 | 0.93 | 0.232 | | 32 A100× N Days |
| SDXL-Turbo [64] | 1 | UNet | 2.6 B | 5.33 | 0.78 | 0.228 | | M GPU× N Days |
| SDXL-Lightning [35] | 1 | UNet | 2.6 B | 5.34 | 0.54 | 0.223 | | 64 A100× N Days |
| Hyper-SDXL[57] | 1 | UNet | 2.6 B | 5.85 | 1.19 | 0.231 | | 32 A100× N Days |
| PixArt-$\alpha$[7] | 30 | DiT | 610 M | 5.97 | 0.82 | 0.226 | | |
| **SIM-DiT-600M** | 1 | DiT | 610 M | 6.42 | 0.67 | 0.223 | | 4 A100× 2 days |
| PixArt-$\alpha^*$ [7] | 30 | DiT | 610 M | 5.93 | 0.53 | 0.223 | 54.88% | |
| **SIM-DiT-600M**$^*$ | 1 | DiT | 610 M | 5.91 | 0.44 | 0.223 | 45.12% | 4 A100× 2 days |

Table 3: Quantitative comparisons with frontier text-to-image models on COCO-2017 validation dataset. The user preference is the winning rate of our user study on SIM-DiT-600M against 20-step PixelArt-$\alpha$. $*$ means the results evaluated on the SAM-LLaVA-Caption10M dataset, and SIM-DiT-600M means the SIM generator distilled from PixelArt-$\alpha$-600M, excluding those in the T5 text encoder. The distillation cost *M GPU× N Days* means the model did not report the cost.

**Failure Cases of One-step SIM-DiT Model.** Though the SIM-DiT one-step model shows impressive performances, it inevitably has limitations. For instance, we find that the 0.6B SIM-DiT one-step model sometimes fails to generate high-quality tiny human faces and proper human arms and fingers. Besides, the model sometimes generates a wrong number of objects and contents that do not strictly follow the prompts. We believe that scaling up the model size and teacher diffusion models will help to address these issues. Please refer to Figure 4 for visualization of failure cases.

## 5 Conclusion and Future Works

This paper presents a novel diffusion distillation method, the score implicit matching (SIM), which enables to transform pre-trained multi-step diffusion models into one-step generators in a data-free fashion. The theoretical foundations and practical algorithms introduced in this paper can enable more affordable deployment of single-step generators across various domains and applications at scale without compromising the performance of underlying generative models.

Nonetheless, SIM has its limitations that call for further research. First, with the abundance of other powerful pre-trained generative models such as flow-matching models, it is worth exploring to reveal if it is possible to generalize the application of SIM to such a broader family of generative models. Second, even though data-free is an important feature of SIM, incorporating new data in the SIM can further boost the quality of generated images failed by the teacher model. This potential benefit has yet to be explored. We hope this could ease the training of large generative models.

## Acknowledgement

Zhengyang Geng is supported by funding from the Bosch Center for AI. Zico Kolter gratefully acknowledges Bosch's funding for the lab.

We would like to acknowledge constructive suggestions from reviewers and ACs/SACs/PCs of NeurIPS 2024. We acknowledge Dr. Mingyuan Zhou for his constructive suggestions on the representation of our theoretical results. We also acknowledge the authors of Diff-Instruct and Score-identity Distillation for their great contributions to high-quality diffusion distillation Python code. We appreciate the authors of PixelArt-$\alpha$ for making their DiT-based diffusion model public.

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

# A Theory Parts

## A.1 Proof of Theorem 3.1

The proof of Theorem 3.1 is based on the so-called Score-projection identity which was first found in Vincent [70] to bridge denoising score matching and denoising auto-encoders. Later the identity is applied by Zhou et al. [93] for deriving distillation methods based on Fisher divergences. We appreciate the efforts of Zhou et al. [93] and re-write the score-projection identity here without proof. Readers can check Zhou et al. [93] for a complete proof of score-projection identity.

**Theorem A.1** (Score-projection identity). Let $\boldsymbol{u}(\cdot, \theta)$ be a vector-valued function, using the notations of Theorem 3.1, under mild conditions, the identity holds:

$$\mathbb{E}_{\substack{\boldsymbol{x}_0 \sim p_{\theta,0} \\ \boldsymbol{x}_t | \boldsymbol{x}_0 \sim q_t(\boldsymbol{x}_t | \boldsymbol{x}_0)}} \boldsymbol{u}(\boldsymbol{x}_t, \theta)^T \left\{ \boldsymbol{s}_{p_{\theta,t}}(\boldsymbol{x}_t) - \nabla_{\boldsymbol{x}_t} \log q_t(\boldsymbol{x}_t | \boldsymbol{x}_0) \right\} = 0, \quad \forall \theta.$$

Next, we turn to prove the Theorem 3.1.

*Proof.* We prove a more general result. Let $\boldsymbol{u}(\cdot)$ be a vector-valued function, the so-called score-projection identity [93, 70] holds,

$$\mathbb{E}_{\substack{\boldsymbol{x}_0 \sim p_{\theta,0} \\ \boldsymbol{x}_t | \boldsymbol{x}_0 \sim q_t(\boldsymbol{x}_t | \boldsymbol{x}_0)}} \boldsymbol{u}(\boldsymbol{x}_t, \theta)^T \left\{ \boldsymbol{s}_{p_{\theta,t}}(\boldsymbol{x}_t) - \nabla_{\boldsymbol{x}_t} \log q_t(\boldsymbol{x}_t | \boldsymbol{x}_0) \right\} = 0, \quad \forall \theta. \tag{A.1}$$

Taking $\theta$ gradient on both sides of identity (A.1), we have

$$0 = \mathbb{E}_{\substack{\boldsymbol{x}_0 \sim p_{\theta,0} \\ \boldsymbol{x}_t | \boldsymbol{x}_0 \sim q_t(\boldsymbol{x}_t | \boldsymbol{x}_0)}} \frac{\partial}{\partial \boldsymbol{x}_t} \left\{ \boldsymbol{u}(\boldsymbol{x}_t, \theta)^T \left\{ \boldsymbol{s}_{p_{\theta,t}}(\boldsymbol{x}_t) - \nabla_{\boldsymbol{x}_t} \log q_t(\boldsymbol{x}_t | \boldsymbol{x}_0) \right\} \right\} \frac{\partial \boldsymbol{x}_t}{\partial \theta}$$

$$+ \mathbb{E}_{\substack{\boldsymbol{x}_0 \sim p_{\theta,0} \\ \boldsymbol{x}_t | \boldsymbol{x}_0 \sim q_t(\boldsymbol{x}_t | \boldsymbol{x}_0)}} \frac{\partial}{\partial \boldsymbol{x}_0} \left\{ \boldsymbol{u}(\boldsymbol{x}_t, \theta)^T \left\{ - \nabla_{\boldsymbol{x}_t} \log q_t(\boldsymbol{x}_t | \boldsymbol{x}_0) \right\} \right\} \frac{\partial \boldsymbol{x}_0}{\partial \theta} \tag{A.2}$$

$$+ \mathbb{E}_{\substack{\boldsymbol{x}_0 \sim p_{\theta,0} \\ \boldsymbol{x}_t | \boldsymbol{x}_0 \sim q_t(\boldsymbol{x}_t | \boldsymbol{x}_0)}} \boldsymbol{u}(\boldsymbol{x}_t, \theta)^T \frac{\partial}{\partial \theta} \left\{ \boldsymbol{s}_{p_{\theta,t}}(\boldsymbol{x}_t) \right\} + \frac{\partial}{\partial \theta} \boldsymbol{u}(\boldsymbol{x}_t, \theta)^T \boldsymbol{s}_\theta(\boldsymbol{x}_t) \tag{A.3}$$

$$= \mathbb{E}_{\substack{\boldsymbol{x}_0 \sim p_{\theta,0} \\ \boldsymbol{x}_t | \boldsymbol{x}_0 \sim q_t(\boldsymbol{x}_t | \boldsymbol{x}_0)}} \boldsymbol{u}(\boldsymbol{x}_t, \theta)^T \frac{\partial}{\partial \theta} \left\{ \boldsymbol{s}_{p_{\theta,t}}(\boldsymbol{x}_t) \right\} \tag{A.4}$$

$$+ \mathbb{E}_{\substack{\boldsymbol{x}_0 \sim p_{\theta,0} \\ \boldsymbol{x}_t | \boldsymbol{x}_0 \sim q_t(\boldsymbol{x}_t | \boldsymbol{x}_0)}} \left\{ \frac{\partial}{\partial \boldsymbol{x}_t} \left\{ \boldsymbol{u}(\boldsymbol{x}_t, \theta)^T \left\{ \boldsymbol{s}_{p_{\theta,t}}(\boldsymbol{x}_t) - \nabla_{\boldsymbol{x}_t} \log q_t(\boldsymbol{x}_t | \boldsymbol{x}_0) \right\} \right\} \frac{\partial \boldsymbol{x}_t}{\partial \theta} \right. \tag{A.5}$$

$$+ \frac{\partial}{\partial \boldsymbol{x}_0} \left\{ \boldsymbol{u}(\boldsymbol{x}_t, \theta)^T \left\{ - \nabla_{\boldsymbol{x}_t} \log q_t(\boldsymbol{x}_t | \boldsymbol{x}_0) \right\} \right\} \frac{\partial \boldsymbol{x}_0}{\partial \theta} \tag{A.6}$$

$$\left. + \frac{\partial}{\partial \theta} \boldsymbol{u}(\boldsymbol{x}_t, \theta)^T \boldsymbol{s}_\theta(\boldsymbol{x}_t) \right\} \tag{A.7}$$

$$= \mathbb{E}_{\boldsymbol{x}_t \sim p_{\theta,t}} \boldsymbol{u}(\boldsymbol{x}_t, \theta)^T \frac{\partial}{\partial \theta} \left\{ \boldsymbol{s}_{p_{\theta,t}}(\boldsymbol{x}_t) \right\} \tag{A.8}$$

$$+ \frac{\partial}{\partial \theta} \mathbb{E}_{\substack{\boldsymbol{x}_0 \sim p_{\theta,0} \\ \boldsymbol{x}_t | \boldsymbol{x}_0 \sim q_t(\boldsymbol{x}_t | \boldsymbol{x}_0)}} \boldsymbol{u}(\boldsymbol{x}_t, \theta)^T \left\{ \boldsymbol{s}_{p_{[\theta],t}}(\boldsymbol{x}_t) - \nabla_{\boldsymbol{x}_t} \log q_t(\boldsymbol{x}_t | \boldsymbol{x}_0) \right\} \tag{A.9}$$

Therefore we have the following identity:

$$\mathbb{E}_{\boldsymbol{x}_t \sim p_{\theta,t}} \boldsymbol{u}(\boldsymbol{x}_t, \theta)^T \frac{\partial}{\partial \theta} \left\{ \boldsymbol{s}_{p_{\theta,t}}(\boldsymbol{x}_t) \right\} = -\frac{\partial}{\partial \theta} \mathbb{E}_{\substack{\boldsymbol{x}_0 \sim p_{\theta,0} \\ \boldsymbol{x}_t | \boldsymbol{x}_0 \sim q_t(\boldsymbol{x}_t | \boldsymbol{x}_0)}} \boldsymbol{u}(\boldsymbol{x}_t, \theta)^T \left\{ \boldsymbol{s}_{p_{[\theta],t}}(\boldsymbol{x}_t) - \nabla_{\boldsymbol{x}_t} \log q_t(\boldsymbol{x}_t | \boldsymbol{x}_0) \right\} \tag{A.10}$$

which holds for arbitrary function $\boldsymbol{u}(\cdot, \theta)$ and parameter $\theta$. If we set

$$\boldsymbol{u}(\boldsymbol{x}_t, \theta) = \mathbf{d}'(\boldsymbol{y}_t)$$
$$\boldsymbol{y}_t = \boldsymbol{s}_{p_{\mathrm{sg}[\theta],t}}(\boldsymbol{x}_t) - \boldsymbol{s}_{q_t}(\boldsymbol{x}_t)$$

Then we formally have

$$
\frac{\partial}{\partial \theta} \mathbb{E}_{\boldsymbol{x}_t \sim p_{\mathrm{sg}[\theta],t}} \left\{ \mathbf{d}'(\boldsymbol{y}_t) \right\}^T \left\{ \boldsymbol{s}_{p_{\theta,t}}(\boldsymbol{x}_t) \right\}
$$
$$
= \frac{\partial}{\partial \theta} \mathbb{E}_{\substack{\boldsymbol{x}_0 \sim p_{\theta,0}, \\ \boldsymbol{x}_t | \boldsymbol{x}_0 \sim q_t(\boldsymbol{x}_t | \boldsymbol{x}_0)}} \left\{ -\mathbf{d}'(\boldsymbol{y}_t) \right\}^T \left\{ \boldsymbol{s}_{p_{\theta,t}}(\boldsymbol{x}_t) - \nabla_{\boldsymbol{x}_t} \log q_t(\boldsymbol{x}_t | \boldsymbol{x}_0) \right\} \tag{A.11}
$$

$\square$

## A.2   Pytorch style pseudo-code of Score Implicit Matching

In this section, we give a PyTorch style pseudo-code for algorithm 1, with the Pseudo-Huber distance function. For a detailed algorithm on CIFAR10 with EDM model, please check Algorithm 2.

```python
import torch
import torch.nn as nn
import torch.optim as optim

# Initialize generator G
G = Generator()

## load teacher DM
Sd = DiffusionModel().load('/path_to_ckpt').eval().requires_grad_(False)
Sg = copy.deepcopy(Sd) ## initialize online DM with teacher DM

# Define optimizers
opt_G = optim.Adam(G.parameters(), lr=0.001, betas=(0.0, 0.999))
opt_Sg = optim.Adam(Sg.parameters(), lr=0.001, betas=(0.0, 0.999))

# Training loop
while True:
    ## update Sg
    Sg.train().requires_grad_(True)
    G.eval().requires_grad_(False)

    # loop for 2 times to update Sg
    for _ in range(2):
      z = torch.randn((2000, 2)).to(device)
      with torch.no_grad():
        fake_x = G(z)

      t = torch.from_numpy(np.random.choice(np.arange(1,Sd.T), size=
    fake_x.shape[0], replace=True)).to(device).long()
      fake_xt, t, noise, sigma_t, g2_t = Sd(fake_x, t=t, return_t=True)
      sigma_t = sigma_t.view(-1,1).to(device)
      g2_t = g2_t.to(device)
      score = Sg(torch.cat([fake_xt,t.view(-1,1)/Sd.T],-1))/sigma_t

      batch_sg_loss = score + noise/sigma_t
      batch_sg_loss = (g2_t*batch_sg_loss.square().sum(-1)).mean()*Sd.T

      optimizer_Sg.zero_grad()
      batch_sg_loss.backward()
      optimizer_Sg.step()

    ## update G
    Sg.eval().requires_grad_(False)
    G.train().requires_grad_(True)

    z = torch.randn((2000, 2)).to(device)
    fake_x = G(z)
```

```
49    t = torch.from_numpy(np.random.choice(np.arange(1,diffusion.T), size=
      fake_x.shape[0], replace=True)).to(device).long()
50    fake_xt, t, noise, sigma_t, g2_t = diffusion(fake_x, t=t, return_t=
      True)
51    sigma_t = sigma_t.view(-1,1).to(device)
52    g2_t = g2_t.to(device)
53
54    score_true = Sd(torch.cat([fake_xt,t.view(-1,1)/diffusion.T],-1))/
      sigma_t
55    score_fake = Sg(torch.cat([fake_xt,t.view(-1,1)/diffusion.T],-1))/
      sigma_t
56
57    score_diff = score_true - score_fake
58
59    offset_coeff = denoise_diff / torch.sqrt(denoise_diff.square().sum
      ([1,2,3], keepdims=True) + self.phuber_c**2)
60    weight = 1.0
61
62    batch_g_loss = weight * offset_coeff * (fake_denoise - images)
63    batch_g_loss = batch_g_loss.sum([1,2,3]).mean()
64
65    optimizer_G.zero_grad()
66    batch_g_loss.backward()
67    optimizer_G.step()
```

Listing 1: Pytorch Style Pseudo-code of SIM

### A.3  Instances of SIM with different distance functions

In section 3.3, we have discussed the powered normed as distance functions. Other choices, such as the Huber distance, which is defined as

$$\forall 1 \leq d \leq D, \ \ L_\delta(\boldsymbol{y})_d := \begin{cases} y_d^2/2 & \text{for } y_d \geq \delta \\ \delta(|y_d| - \delta/2) & \text{otherwise} \end{cases}$$

For other choices of distance functions, such as $L1$ norm and exponential with powered norms, we put them in Table 4.

Table 4: Instances of Score Implicit Matching loss with different distance functions. The notations are aligned with the Algorithm 1.

| CHOICE OF $\mathbf{d}(.)$ | $\mathbf{d}'(\boldsymbol{y}_t)$ | LOSS FUNCTION |
|---|---|---|
| $\|\boldsymbol{y}_t\|_2^2$ | $2\boldsymbol{y}_t$ | $-2\boldsymbol{y}_t^T\left\{\boldsymbol{s}_\psi(\boldsymbol{x}_t,t) - \nabla_{\boldsymbol{x}_t}\log q_t(\boldsymbol{x}_t\|\boldsymbol{x}_0)\right\}$ |
| $\|\boldsymbol{y}_t\|_\alpha^\alpha, \ \alpha\geq 1,$ $\alpha\ even$ | $\alpha\boldsymbol{y}_t^{(\alpha-1)}$ | $-\alpha\left\{\boldsymbol{y}_t^{(\alpha-1)}\right\}^T\left\{\boldsymbol{s}_\psi(\boldsymbol{x}_t,t) - \nabla_{\boldsymbol{x}_t}\log q_t(\boldsymbol{x}_t\|\boldsymbol{x}_0)\right\}$ |
| $\exp(\beta\|\boldsymbol{y}_t\|_\alpha^\alpha)-1,$ $\alpha\geq 1, \alpha\ even$ | $\alpha\exp(\beta\|\boldsymbol{y}_t\|_\alpha^\alpha)\boldsymbol{y}_t^{(\alpha-1)}$ | $-\alpha\exp(\beta\|\boldsymbol{y}_t\|_\alpha^\alpha)\left\{\boldsymbol{y}_t^{(\alpha-1)}\right\}\left\{\boldsymbol{s}_\psi(\boldsymbol{x}_t,t) - \nabla_{\boldsymbol{x}_t}\log q_t(\boldsymbol{x}_t\|\boldsymbol{x}_0)\right\}$ |
| $\|\boldsymbol{y}_t\|_1$ | $\text{sign}(\boldsymbol{y}_t)$ | $-\text{sign}(\boldsymbol{y}_t)^T\left\{\boldsymbol{s}_\psi(\boldsymbol{x}_t,t) - \nabla_{\boldsymbol{x}_t}\log q_t(\boldsymbol{x}_t\|\boldsymbol{x}_0)\right\}$ |
| $L_\delta(\boldsymbol{y}_t),$ $L_\delta(.)\ \text{HUBER LOSS}$ | $\frac{\partial}{\partial\boldsymbol{y}_t}L_\delta(\boldsymbol{y}_t)$ | $-\frac{\partial}{\partial\boldsymbol{y}_t}L_\delta(\boldsymbol{y}_t)^T\left\{\boldsymbol{s}_\psi(\boldsymbol{x}_t,t) - \nabla_{\boldsymbol{x}_t}\log q_t(\boldsymbol{x}_t\|\boldsymbol{x}_0)\right\}$ |
| $\sqrt{\|\boldsymbol{y}_t\|_2^2 + c^2} - c$ | $2\frac{\boldsymbol{y}_t}{\sqrt{\|\boldsymbol{y}_t\|_2^2+c^2}}$ | $-2\left\{2\frac{\boldsymbol{y}_t}{\sqrt{\|\boldsymbol{y}_t\|_2^2+c^2}}\right\}^T\left\{\boldsymbol{s}_\psi(\boldsymbol{x}_t,t) - \nabla_{\boldsymbol{x}_t}\log q_t(\boldsymbol{x}_t\|\boldsymbol{x}_0)\right\}$ |

## B  Empirical Parts

### B.1  Answer for the human preference study

The answer to the human preference study in Figure 1 is

- the middle image of the first row is generated by one-step SIM-DiT-600M;

- the leftmost image of the second row is generated by one step SIM-DiT-600M;
- the leftmost image of the third row is generated by one-step SIM-DiT-600M.

## B.2 Experiment details on CIFAR10 dataset

We follow the experiment setting of SiD and DI on CIFAR10. We start with a brief introduction to the EDM model [26].

The EDM model depends on the diffusion process

$$\mathrm{d}\boldsymbol{x}_t = t\mathrm{d}\boldsymbol{w}_t, t \in [0, T]. \tag{B.1}$$

Samples from the forward process (B.1) can be generated by adding random noise to the output of the generator function, i.e., $\boldsymbol{x}_t = \boldsymbol{x}_0 + t\boldsymbol{\epsilon}$ where $\boldsymbol{\epsilon} \sim \mathcal{N}(\boldsymbol{0}, \boldsymbol{I})$ is a Gaussian vector. The EDM model also reformulates the diffusion model's score matching objective as a denoising regression objective, which writes,

$$\mathcal{L}(\psi) = \int_{t=0}^{T} \lambda(t) \mathbb{E}_{\boldsymbol{x}_0 \sim p_0, \boldsymbol{x}_t | \boldsymbol{x}_0 \sim p_t(\boldsymbol{x}_t | \boldsymbol{x}_0)} \|\boldsymbol{d}_\psi(\boldsymbol{x}_t, t) - \boldsymbol{x}_0\|_2^2 \mathrm{d}t. \tag{B.2}$$

Where $\boldsymbol{d}_\psi(\cdot)$ is a denoiser network that tries to predict the clean sample by taking noisy samples as inputs. Minimizing the loss (B.2) leads to a trained denoiser, which has a simple relation to the marginal score functions as:

$$\boldsymbol{s}_\psi(\boldsymbol{x}_t, t) = \frac{\boldsymbol{d}_\psi(\boldsymbol{x}_t, t) - \boldsymbol{x}_t}{t^2} \tag{B.3}$$

Under such a formulation, we actually have pre-trained denoiser models for experiments. Therefore, we use the EDM notations in later parts.

**Construction of the one-step generator.** Let $\boldsymbol{d}_\theta(\cdot)$ be pretrained EDM denoiser models. Owing to the denoiser formulation of the EDM model, we construct the generator to have the same architecture as the pre-trained EDM denoiser with a pre-selected index $t^*$, which writes

$$\boldsymbol{x}_0 = g_\theta(\boldsymbol{z}) := \boldsymbol{d}(\boldsymbol{z}, t^*), \quad \boldsymbol{z} \sim \mathcal{N}(\boldsymbol{0}, (t^*)^2 \boldsymbol{I}). \tag{B.4}$$

We initialize the generator with the same parameter as the teacher EDM denoiser model.

**Time index distribution.** When training both the EDM diffusion model and the generator, we need to randomly select a time $t$ in order to approximate the integral of the loss function (B.2). The EDM model has a default choice of $t$ distribution as log-normal when training the diffusion (denoiser) model, i.e.

$$t \sim p_{EDM}(t): \quad t = \exp(s) \tag{B.5}$$

$$s \sim \mathcal{N}(P_{mean}, P_{std}^2), \quad P_{mean} = -1.2, P_{std} = 1.2. \tag{B.6}$$

And a weighting function

$$\lambda_{EDM}(t) = \frac{(t^2 + \sigma_{data}^2)}{(t \times \sigma_{data})^2}. \tag{B.7}$$

In our algorithm, we follow the same setting as the EDM model when updating the online diffusion (denoiser) model.

In SiD, they propose to use a special discrete time distribution, which writes

$$\sigma_k = (\sigma_{max}^{\frac{1}{\rho}} \frac{i}{K-1} (\sigma_{min}^{\frac{1}{\rho}} - \sigma_{max}^{\frac{1}{\rho}}))^\rho,$$

$$\sigma_{max} = 80.0, \sigma_{min} = 0.002, \rho = 7.0, K = 1000$$

They proposed to choose $t$ uniformly from

$$t \sim p_{SiD}(t): \quad k \sim Unif[0, 800], t = \sigma_k; \tag{B.8}$$

We name such a time distribution the $Karr$ distribution in Figure 2 because such a schedule was originally proposed in Karras' EDM work for sampling.

Table 5: Hyperparameters used for SIM on CIFAR10 EDM Distillation

| Hyperparameter | CIFAR-10 (Uncond) | | CIFAR-10 (Cond) | |
|---|---|---|---|---|
| | DM $s_\psi$ | Generator $g_\theta$ | DM $s_\psi$ | Generator $g_\theta$ |
| Learning rate | 1e-5 | 1e-5 | 1e-5 | 1e-5 |
| Batch size | 256 | 256 | 256 | 256 |
| $\sigma(t^*)$ | 2.5 | 2.5 | 2.5 | 2.5 |
| $Adam\ \beta_0$ | 0.0 | 0.0 | 0.0 | 0.0 |
| $Adam\ \beta_1$ | 0.999 | 0.999 | 0.999 | 0.999 |
| Time Distribution | $p_{EDM}(t)$(B.5) | $p_{SIM}(t)$(B.9) | $p_{EDM}(t)$(B.5) | $p_{SIM}(t)$(B.9) |
| Weighting | $\lambda_{EDM}(t)$(B.7) | 1 | $\lambda_{EDM}(t)$(B.7) | 1 |
| Loss function | (B.2) | (B.13) | (B.2) | (B.13) |
| Number of GPUs | 4×A100-40G | 4×A100-40G | 4×A100-40G | 4×A100-40G |

However, in practice, we find that $Karr$ distribution (B.8) empirically does not work well. Instead, we find that a modified log-normal time distribution when updating the generation with SIM works better than $Karr$ distribution. Our SIM time distribution writes:

$$t \sim p_{SIM}(t): \quad t = \exp(s) \tag{B.9}$$

$$s \sim \mathcal{N}(P_{mean}, P_{std}^2), \quad P_{mean} = -3.5, P_{std} = 2.5. \tag{B.10}$$

**Weighting function.** As we have said, we use the same $\lambda_{EDM}(t)$ (B.7) weighting function as EDM when updating the denoiser model. When updating the generator, SiD uses a specially designed weighting function, which writes:

$$w_{SiD}(t) = \frac{C \times t^4}{\|\boldsymbol{x}_0 - \boldsymbol{d}_{q_t}(\boldsymbol{x}_t)\|_{1,\text{sg}}} \tag{B.11}$$

$$\boldsymbol{x}_t = \boldsymbol{x}_0 + t\epsilon, \quad \epsilon \sim \mathcal{N}(\boldsymbol{0}, \mathbf{I}) \tag{B.12}$$

The notation sg means stop-gradient, and $C$ is the data dimensions. They claim such a weighting function helps to stabilize the training. However, in our experiments, since the SIM itself has normalized the loss (see section 4), we do not use such ad-hoc weighting functions. Instead, we just set the weighting function to be 1 for all time. We call the SiD's weighting function the $sidwgt$ in Figure 2, and our weighting the $nowgt$ in Figure 2.

In Figure 2, we compare the SiD and SIM with different time distribution and weighting functions. We find that SIM+nowgt+lognormal time distribution gives the best performances significantly, therefore our final experiment tasks such a configuration. Table 5 records the detailed configurations we use for SIM on CIFAR10 EDM distillation.

With the optimal setting and EDM formulation, we can rewrite our algorithm in an EDM style in Algorithm 2.

## B.3 Experiment details on Text-to-Image Distillation

In the Text-to-Image distillation part, in order to align our experiment with that on CIFAR10, we rewrite the PixArt-$\alpha$ model in EDM formulation:

$$\boldsymbol{d}(\boldsymbol{x};t) = \boldsymbol{x} - tF_\theta \tag{B.14}$$

Here, following the iDDPM+DDIM preconditioning in EDM, PixArt-$\alpha$ is denoted by $F_\theta$, $\boldsymbol{x}$ is the image data plus noise with a standard deviation of $t$, for the remaining parameters such as $C_1$ and $C_2$, we kept them unchanged to match those defined in EDM. Unlike the original model, we only retained the image channels for the output of this model. Since we employed the preconditioning of iDDPM+DDIM in the EDM, each $\sigma$ value is rounded to the nearest 1000 bins after being passed into the model. For the actual values used in PixArt-$\alpha$, beta_start is set to 0.0001, and beta_end is set to 0.02. Therefore, according to the formulation of EDM, the range of our noise distribution is [0.01, 156.6155], which will be used to truncate our sampled $t$. For our one-step generator, it is formulated as:

$$g_\theta(\boldsymbol{z}) = \boldsymbol{d}(\boldsymbol{z}, t^*) = \boldsymbol{z} - t^*F_\theta \tag{B.15}$$

Here following SiD $t^* = 2.5$ and $\boldsymbol{z} \sim \mathcal{N}(0, (t^*)^2\mathbf{I})$, we observed in practice that larger values of $t^*$ lead to faster convergence of the model, but the difference in convergence speed is negligible for the complete model training process and has minimal impact on the final results.

**Algorithm 2:** SIM with Pseudo-Huber distance for distilling EDM teacher [Pytorch Style].

**Input:** pre-trained EDM denoiser $d_{q_t}(.)$, generator $g_\theta$, prior distribution $p_z$, online EDM
denoiser $d_\psi(.)$; differentiable distance function $\mathbf{d}(.)$, and forward diffusion (2.1).

**while** *not converge* **do**

    //      *freeze $\theta$, update $\psi$:*

    $\boldsymbol{x}_0 = g_\theta(\boldsymbol{z}).detach(), \;\; \boldsymbol{z} \sim p_z$

    $t \sim p_{EDM}(t), \;\; \boldsymbol{x}_t = \boldsymbol{x}_0 + t\epsilon, \;\; \epsilon \sim \mathcal{N}(\mathbf{0}, \mathbf{I})$

    $\mathcal{L}(\psi) = \lambda_{EDM}(t) \times \|\boldsymbol{d}_\psi(\boldsymbol{x}_t, t) - \boldsymbol{x}_0\|_2^2$

    $\mathcal{L}(\psi).backward(); \;\; \text{update} \;\; \psi$

    //      *freeze $\psi$, update $\theta$:*

    $\boldsymbol{x}_0 = g_\theta(\boldsymbol{z}), \;\; \boldsymbol{z} \sim p_z$

    $t \sim p_{SIM}(t), \;\; \boldsymbol{x}_t = \boldsymbol{x}_0 + t\epsilon, \;\; \epsilon \sim \mathcal{N}(\mathbf{0}, \mathbf{I})$

$$\mathcal{L}(\theta) = -\left\{\frac{\boldsymbol{y}_t}{\sqrt{\|\boldsymbol{y}_t\|_2^2 + c^2}}\right\}^T \left\{\boldsymbol{d}_\psi(\boldsymbol{x}_t, t) - \boldsymbol{x}_0\right\}, \;\; \text{where } \boldsymbol{y}_t := \boldsymbol{d}_\psi(\boldsymbol{x}_t, t) - \boldsymbol{d}_{q_t}(\boldsymbol{x}_t)$$

$$(B.13)$$

    $\mathcal{L}(\theta).backward(); \;\; \text{update} \;\; \theta$

**end**

**return** $\theta, \psi$.

---

We utilized the SAM-LLaVA-Caption10M dataset, which comprises prompts generated by the LLaVA model on the SAM dataset. These prompts provide detailed descriptions for the images, thereby offering us a challenging set of samples for our distillation experiments.

All experiments in this section were conducted on 4 A100-40G GPUs with bfloat16 precision, using the PixArt-XL-2-512x512 model version, employing the same hyperparameters. For both optimizers, we utilized Adam with a learning rate of 5e-6 and betas=[0, 0.999]. Additionally, to enable a batch size of 1024, we employed gradient checkpointing and set the gradient accumulation to 8. Finally, regarding the training noise distribution, instead of adhering to the original iDDPM schedule, we sample the $\sigma$ from a log-normal distribution with a mean of -2.0 and a standard deviation of 2.0, we use the same noise distribution for both optimization steps and set the two loss weighting to constant 1. Our best model was trained on the SAM Caption dataset for approximately 16k iterations, which is equivalent to less than 2 epochs. This training process took about 2 days on 4 A100-40G GPUs.

We also tested the impact of different noise distributions on the distillation process. When the noise distribution is highly concentrated around smaller values, we observed a phenomenon where the generated samples appear excessively dark. On the other hand, when we used slightly larger noise distributions, we found that the structure of the generated samples tended to be unstable.

### B.4   Instruction for Human Preference Study

Our user study primarily focuses on comparing the outputs of the distilled model and the teacher model. Each image has undergone rigorous manual review to ensure the safety of survey participants. We conducted the study using questionnaires, where users were presented with two randomly ordered images generated by the distilled model and teacher model and asked to select the sample that best matched the text description and had higher image quality. Finally, we used the collected votes for the distilled model and the teacher model as indicators of user preference. The questionnaire website used for conducting these evaluations are shown in Figure 5.

To be more specific, we randomly selected 17 prompt words and generated images of resolution 512x512 using both the student model and the teacher model. To facilitate comparison, we presented the two images side by side in random order. In the questionnaire, we provided the complete prompt words for reference in addition to the generated images. In the end, we collected approximately 30 survey responses in total.

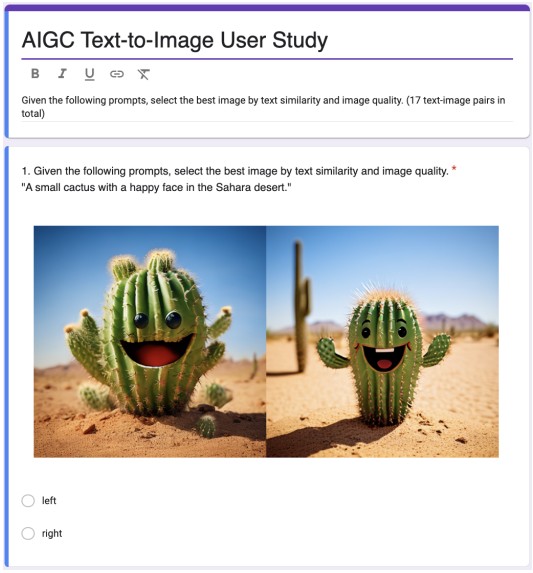

Figure 5: Demonstration of our human preference user study interface.

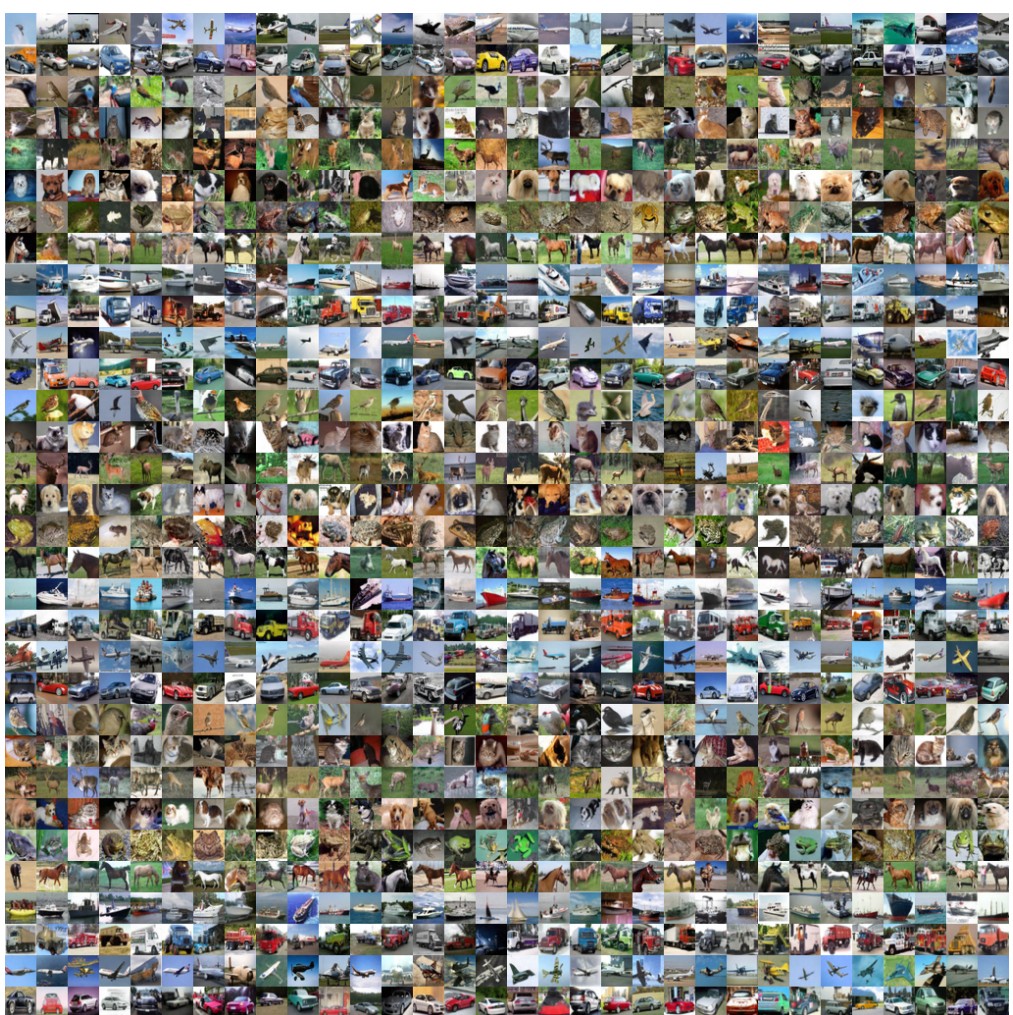

Figure 6: One-step SIM model on CIFAR10-conditional. FID=1.96.

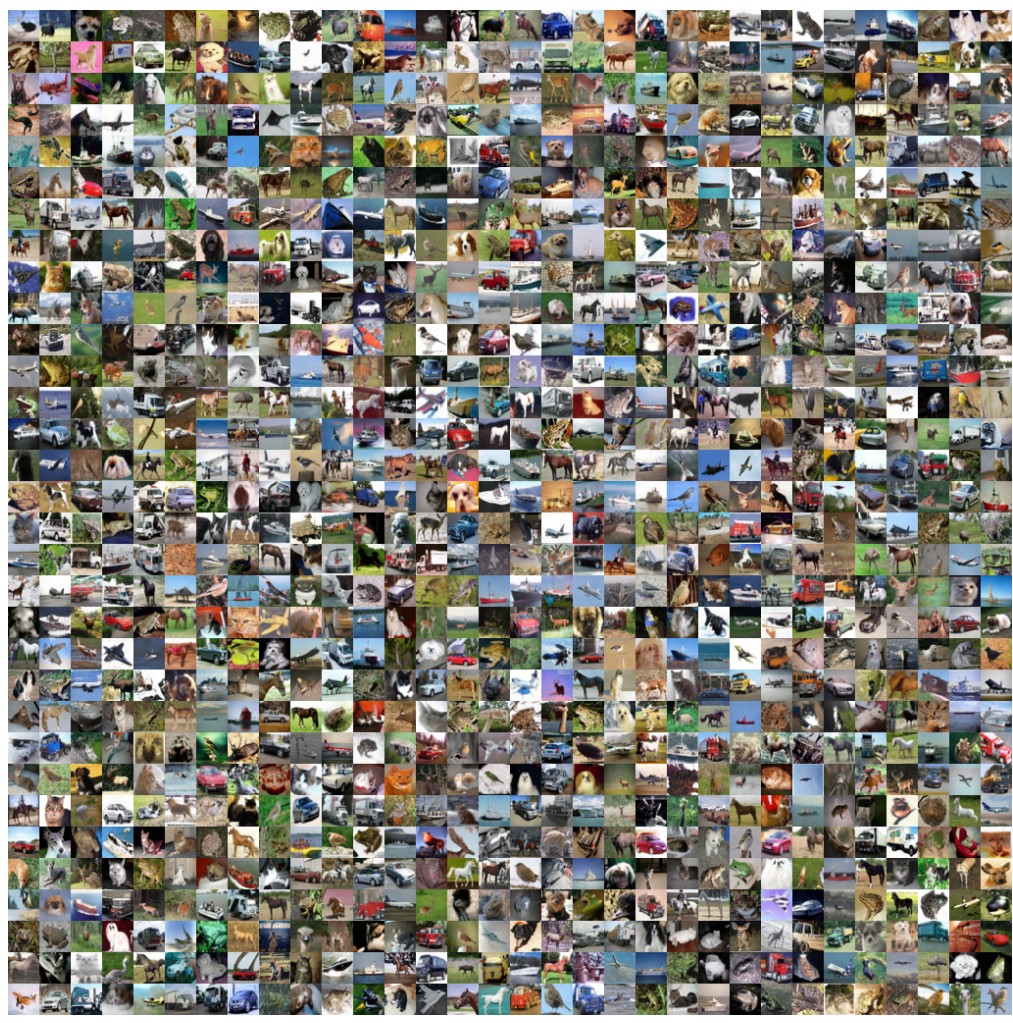

Figure 7: One-step SIM model on CIFAR10-unconditional. FID=2.06.

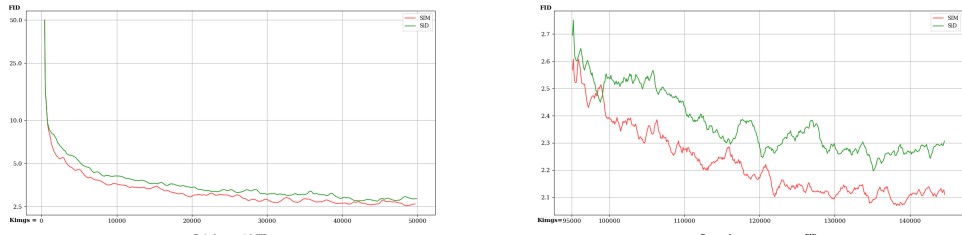

Figure 8: The comparison of FID convergence between SIM and SiD.

## B.5 Generated Samples on CIFAR10

## B.6 FID Convergence on CIFAR10 Unconditional Generation

## B.7 Prompts for Figure 3

- prompt for first row of Figure 3: *A small cactus with a happy face in the Sahara desert.*
- prompt for second row of Figure 3: *An image of a jade green and gold coloured Fabergé egg, 16k resolution, highly detailed, product photography, trending on artstation, sharp*

*focus, studio photo, intricate details, fairly dark background, perfect lighting, perfect composition, sharp features, Miki Asai Macro photography, close-up, hyper detailed, trending on artstation, sharp focus, studio photo, intricate details, highly detailed, by greg rutkowski.*

- prompt for third row of Figure 3: *Baby playing with toys in the snow.*

