# OpenReview forum: "One-Step Diffusion Distillation through Score Implicit Matching"
_NeurIPS.cc/2024/Conference — NeurIPS 2024 poster_

### Official Review · Reviewer_WiWu · 2024-06-17

**Soundness:** 3
**Presentation:** 3
**Contribution:** 3
**Rating:** 7
**Confidence:** 4

**Summary:**

This paper proposes a distillation-based accelerated sampling method for various score-based diffusion models, such as EDM and Stable Diffusion. The authors have named this method Score Implicit Matching (SIM), which is designed to compress information from a diffusion-based teacher into single-step generator models. This method is somewhat based on SID, and SIM can be regarded as an extended version of SID to some extent. In my opinion, the crucial contributions of this paper are its application of SIM to Stable Diffusion and the more general design choice of the distance function.

**Strengths:**

1. This paper applies SIM to text-to-image (t2i) generators such as Stable Diffusion, an important experiment validating the effectiveness of accelerated sampling algorithms.

2. This paper enhances SID by generalizing the design choice of the distance function to six different versions, as shown in Table 4 of the Appendix.

3. This paper provides proof of their method, although the contribution of this point is relatively small since most of the proofs are based on previous work. To be specific, (1) the proof in SID use the data distribution as the target but SIM utilize the score function as target; (2) SIM uses the derivative form for derivation, but SID draws its conclusions through
$\nabla_{x_t}\log p_\theta(x_t)=[x_g-x_t]/\sigma^2_t$
 (In EDM $\alpha_t \equiv 1$); (3) the only difference is the difference in the distance function, but this does not affect the derivation in any way.

**Weaknesses:**

1. The performance of SIM seems to be not effective, although the authors' statement that SIDs are reproduced because they don't have publicly available code. The performance gap may originate from a few tricks, and it is not yet known whether these tricks can further improve the effectiveness of SIM. Thus. I hope the authors can retrain several experiments related to CIFAR10 to further evaluate the effectiveness of SIM through SID's official implementation [1].

[1] https://github.com/mingyuanzhou/sid

2. The author spends a lot of space in the ``Score Implicit Matching" section to introduce the principle of Algorithm 1, but I think the biggest difference between SIM and SID is that SIM adopts the matching based on the score function while SID adopts the matching based on the sample. The rest of the logic is basically the same. Therefore, I think it is a bit unreasonable to take this part as the contribution of the paper.

3. The authors of SID recently released a new paper named ``Long and Short Guidance in Score identity Distillation for One-Step Text-to-Image Generation'' (LSG) which is also an extended version of SID. I wish the authors would add LSG into related work (unfortunately, different metrics were used between SIM and LSG, so SIM cannot simply compare it with LSG).

**Questions:**

No

**Limitations:**

Yes

---

> ### Author Rebuttal · Authors · 2024-08-07
>
> Thank you for your useful feedback. We will address your concerns one by one. Before that, we first give a summary of our work.
>
> In this work, we introduce score implicit matching (SIM), a novel distillation algorithm that achieves competitive performances on CIFAR10 generation tasks, and **lossless aesthetic performances** on the one-step distillation of the **DiT-based text-to-image diffusion model**. Besides the strong empirical performances, the theoretical foundation of SIM is also sound: we prove that the **SIM loss shares the same $\theta$ gradient as our proposed score-based divergence**. Therefore using an SGD-type optimization algorithm to minimize SIM loss is secretly equivalent to minimizing the intractable score-based divergence. This **distinguishes SIM from SiD's theoretical loss functions** that are proved to be equal to Fisher divergence in value instead of parameter gradient when using an online learned score to replace the unknown generator score.
>
> Next, we address your concerns one by one.
>
> **Q1**. The relation between SIM and SiD
>
> **A1**. We appreciate the SiD paper and have obtained inspiration from it. The SiD shows a solid step to minimize Fisher divergence other than KL. However, SIM and SiD are essentially different in the following aspects.
>
> **1)** SIM loss has the same parameter gradient as the intractable score-based divergence. However, the SiD's theoretical losses only have equal value instead of an equal parameter gradient.
>
> In Section 3.2, we have shown that SIM loss has the same parameter gradient (equation 3.5-3.7) as the target score-based divergence (equation 3.4) regardless of substituting the score function with an online learned score model (please check Theorem 3.1). This means that using an SGD-type optimization algorithm to minimize SIM loss is equivalent to minimizing the score-based divergence. On the contrary, the SiD theoretical loss functions (equation 13 and equation 20 in SiD paper) only have an equal value to the target Fisher divergence instead of the parameter gradient when using an online score function to replace the intractable generator score function. This means using a gradient-based optimization algorithm does not guarantee the minimization of the Fisher divergence. This might be the reason why SiD's theoretical-driven losses do not work well in practice as the authors of SiD have pointed out. After the theoretical argument, the authors of SiD empirically find that a linear combination of the two "bad" losses results in a strong loss function, which they call the "fused loss". Interestingly, we find that when taking SIM's distance function to be squared L2 distance, SIM recovers the empirical "fused loss" of SiD with $\alpha=1.0$. **From this point of view, SIM, as well as our score-divergence gradient theorem brings a theoretical tool for analyzing SiD's empirical fused loss in a gradient view.**
>
> **2)** SIM is defined over a general family of score-based divergence with a flexible choice of distance function. As a comparison, the SiD only targets Fisher divergence, which can be viewed as a special case of SIM. Besides, trying to generalize SiD's theoretical loss (equation 20 in SiD) to support general distance functions is not direct. The proof of Equation 19 of the SiD paper relies on the expansion of the squared L2 difference. However, such an expansion may not hold for general distance functions such as L2 distance and the Pseudo-Huber distance.
>
> **3)** SIM empirically shows faster convergence than SiD. This makes SIM favored when training large-scale one-step text-to-image generator models. Please see **A2** for details.
>
> **4)** SIM shows novel scalability to complicated text-to-image one-step distillation, however, the SiD did not show such a possibility before the NeurIPS submission.
>
> In short conclusion, SIM has proven the equality of the loss function and the intractable divergence in a **parameter gradient** view, making it a solid theoretical contribution. Besides, SIM has shown success in scaling to train strong one-step text-to-image generators without performance losses, making it a solid empirical contribution.
>
> **Q2**. Will SIM be further improved with the SiD codebase?
>
> **A2**. We appreciate your constructive suggestion. After reading the SiD's official code, we find the official code of SiD has multiple tricks that are different from our original implementation. With some tests, we find that these tricks, such as masking out the nan samples, the weighting function, as well as the continuous-time Karras-$\rho$ noise sigma distribution help stabilize the training and improve the performance.
>
> Due to the limited time of the rebuttal period and limited compute, we conduct experiments with two settings to compare SIM and SiD behavior at different training stages: **(1) training from scratch**, and **(2) resuming from a nearly converged** generator; **Please check our global rebuttal and added one-page** for technical details.
>
> The new experiment shows that the faster convergence of SIM than SiD ($\alpha=1.0$) using the same implementation techniques. However, training to the end will take a very long time, and we have to acknowledge that we probably can not make a final conclusion that SIM will be better than SiD without sufficient computation. Besides, we appreciate SiD's high-quality implementations and solid contributions.
>
> **Q3**. The authors of SID recently released a new paper named LSG, I wish the authors would add LSG into related work.
>
> **A3**. Thank you for your kind reminder. The LSG is publicly available after the NeurIPS submission deadline. After a careful reading, we find it technically interesting. We think the Long-and-short guidance technique proposed in LSG may further improve the SIM performance for text-to-image models. We will add the LSG to our related work in the revision.
>
> **We hope our response has resolved your concerns. If you still have any concerns, please let us know.**

---

> > ### Comment · Reviewer_WiWu · 2024-08-11
> >
> > I acknowledge the authors' response and appreciate their efforts in addressing all my concerns. I believe this paper is now well-prepared for acceptance.

---

> > > ### Author Response · Authors · 2024-08-11
> > > **Thank you for your constructive feedback.**
> > >
> > > We are glad that we have addressed your concerns. We appreciate your valuable suggestions and will incorporate them in our revision.

---

### Official Review · Reviewer_Jq2C · 2024-07-11

**Soundness:** 3
**Presentation:** 3
**Contribution:** 3
**Rating:** 8
**Confidence:** 3

**Summary:**

The authors study the distillation of score-based models in one-step generators. They propose a new objective function for score distillation coined Score-Based Divergence. This divergence measures the mean distance between the pre-trained score-based model and a score-based model learned on the distribution induced by the one-step generator. This method is then applied on image generation problems, on standard academic benchmarks and on large latent text-to-image models.

**Strengths:**

Overall, this is a strong paper with a sound and novel method that obtains solid experimental results.

* The authors study the distillation of score-based models, which is a very hot research topic since inference time is the major bottleneck of such models.

* As far as I know, the idea of score-based divergence is novel and original. Moreover, the method comes with state-of-the-art results in a very competitive area (distillation of score-based models to one-step generators).

* The general idea of score-based divergences is sound and could lead to new research works, from applied ones to theoretical ones.

**Weaknesses:**

* Not a big weakness, but we could argue that the training is demanding from a computational viewpoint, since it involves three large neural networks: generator, score-based network for data distribution, score-based network for generator's distribution. However, this is not specific to this particular method.

* Minor: typo in the TL;DR. "The submission propose an general" -> "The submission proposes a general"

**Questions:**

* Have you observed instabilities with the optimization procedure that alternates between online score and generator? Have you tried to explore other hyper-parameters, e.g. more steps or different learning rates for the online score network?

* In ProlificDreamer, the authors use LORA to fine-tune the online score-based model. Have you tried such approach to reduce the computational cost?

* Could your method be directly used for learning 3D models, such as ProlificDreamer? From what I understand, setting $\theta$ to an ensemble of particles and generating 2D images through NERFs, this would be practicable.

* Theoretically, your method is agnostic to the type of generator that is used, right? Thus, we could imagine training a model with low-dimensional latent space and different architecture, such as a StyleGAN. Have you explored this?

**Limitations:**

.

---

> ### Author Rebuttal · Authors · 2024-08-07
>
> We are glad that you like our work. We appreciate your valuable suggestions, and we will incorporate them in our revision. In this paper, we introduce score implicit matching (SIM), a theoretically sound distillation method that secretly minimizes a general family of score-based divergences when distillation. Our experiments on CIFAR10 generation as well as the one-step DiT-based text-to-image generator model show the superior performances of SIM on industry-level applications. In the following paragraphs, we will address your concerns one by one.
>
> **Q1**. Not a big weakness, but the training computation cost may be demanding.
>
> **A1**. We acknowledge that the training cost is a little bit expensive because the SIM involves an additional online diffusion model. However, in practice, we find that the **additional cost is acceptable** by using Pytorch techniques. For example, in our text-to-image experiment, we use 4 Nvidia A100-80G GPUs for training the one-step generator. We use the Pytorch Distributed data-parallel (DDP) to support multi-GPU gradient syncing and use BF16 numerical formate for training. With BF16 training, we can effectively train the model with a total batch size of 256 even without the gradient accumulation. If we use gradient accumulation techniques, we can enlarge the batch size to 1024 with some speed decline.
>
> **Q2**. Are there instabilities with the alternating optimization procedure? Would be better to explore other hyper-parameters.
>
> **A2**. We appreciate your good intuition. We have observed that **if the hyper-parameters are not set properly, the training may diverge**. For instance, we find that when the weighting functions and the log-normal noise sigma distribution for training online diffusion models are not properly set, the alternating training will lead to a poor result. However, compared with Diff-Instruct and SiD, we find that the updating of a one-step generator using SIM is pretty robust to hyper-parameters. We find that using different loss scaling, different learning rates (in a reasonable interval), as well as several kinds of weighting functions, does not affect the SIM updates for the generator too much. We appreciate your constructive suggestions as a new direction to explore. We have tried to update the online diffusion twice, however, this slows down the training process by approximately 1.5 times and we do not observe significant performance improvement, therefore, we still use one diffusion update per generator update, with two updates using the learning learning rate. However, we will explore more of these settings in the revision.
>
> **Q3**. Have you tried using Lora such as in ProlificDreamer [2] to reduce the computational cost?
>
> **A3**. We appreciate your good intuition. We like the trick proposed in ProlificDreamer to use a Lora online diffusion, which can reduce the memory cost. However, as we have shown in **A1**, when we use BF16 training, we are not bothered by memory issues, therefore we do not use the Lora online diffusion. Another reason is that the Lora model may have worse modeling ability than a full-parameter model, which can potentially harm the performance of SIM.
>
>
> **Q4**. Can SIM be used to train NeRF models using 2D diffusion? Can SIM be applied to other generator models such as GANs?
>
> **A4.** We highly appreciate your good intuition. As you can see, the SIM is a general method that can be used for a wide range of applications other than one-step diffusion distillation. In the rebuttal period, we conduct two more experiments: (1) text-to-3D generation using text-to-2D diffusion; and (2) improving the GAN generator using diffusion models; **Please check our Global rebuttal cell for details.** On both new applications, SIM has shown strong performances, demonstrating its broad applicability.
>
> **We hope our response has resolved your concerns. If you still have any concerns, please let us know.**

---

> > ### Comment · Reviewer_Jq2C · 2024-08-09
> > **Rebuttal Acknowledgement**
> >
> > I acknowledge the rebuttal. The authors have thoroughly replied to my questions. I will thus raise my score to 8, Strong Accept.

---

> > > ### Author Response · Authors · 2024-08-11
> > > **Thank you for your constructive feedback.**
> > >
> > > We are glad that we have addressed your concerns. We appreciate your valuable suggestions and will incorporate them in our revision.

---

### Official Review · Reviewer_wPU2 · 2024-07-12

**Soundness:** 3
**Presentation:** 2
**Contribution:** 3
**Rating:** 6
**Confidence:** 4

**Summary:**

This work proposed Score Implicit Matching (SIM) to distill diffusion models into a one-step generator. The core idea is to use the “score-gradient theorem” to transform the minimization of score-based divergences between generator and real score functions into an implicit and tractable minimization problem. The theory does not specify the underlying divergence or distance metric for score functions, and seems to generalize many distribution matching methods, such as SiD and DI. Experiments were conducted on CiFAR-10 and text-to-image generation to show the effectiveness and robustness of SIM, by distilling EDM and PixArt-$\alpha$, respectively.

**Strengths:**

- Overall, the paper is well written and easy to read, although there is still a need to further polish the presentation (as I pointed in the Weaknesses section).
- This work provides a good theoretical insight into how to transform the intractable score-based divergence into a tractable implicit minimization problem, where both share the same gradient.
- It generalizes many previous works (SiD, DI) as its specific cases.
- Experiments on CIFAR-10 and text-to-image generation showed convincing results to demonstrate the effectiveness of SIM.

**Weaknesses:**

- The reason why Pseudo-Huber distance is better than other choices (L2 Square in SiD or KL in DMD) is not quite clear to me. Can the authors provide some empirical evidence to support the hypothesis of the normalization effect? For example, we can use L2 distance (or setting c=0), does the method also work well as it also normalizes the difference between generator score and teacher score? Also, what $c$ value works the best for the Pseudo-Huber distance?
- Can the authors show how SIM becomes a DMD objective if we set $d(\cdot)$ to a reserve KL as in DMD? It is not quite clear to me how it works.
- There are some other standard benchmarks for one-step distillation, including distilling EDM trained on ImageNet-64 and SD trained on text-image pairs. I wonder why not compare SIM with strong baselines on these two settings?
- The presentation can be further improved. For example, the font size in Figure 2 is too small, and the legends in the right two figures are given without any explanation. Also, all the hyperlinks of references and citations are missing throughout the paper. Besides, there is a few typos: In Table 1, the FID of SiD ($\alpha=1$) and SiD ($\alpha=1.2$) can be switched. In line 24, “data synthetic” might be corrected as “data synthesis”. In line 254, “one generation steps” might be corrected as “one-step generation”.
- This is not a major concern but there is an obvious gap between the reproduced SiD results and their reported ones. The authors admitted that it is because SiD code was not released and more importantly, the SIM results can be further improved if its hyperparameters can be tuned based on SiD’s setting. So I wonder that since SiD code has now been released: https://github.com/mingyuanzhou/SiD, maybe it is a good time to look into the reproduction issue and decide whether it can improve SIM’s performance.

**Questions:**

Please see my questions in the Weaknesses section.

**Limitations:**

The authors adequately addressed the limitations and potential negative societal impact of their work.

---

> ### Author Rebuttal · Authors · 2024-08-07
>
> Thank you for your useful suggestions. We will address your concerns one by one. Before that, we first give a summary of the main contributions of our work.
>
> In this work, we introduce score implicit matching (SIM), a novel distillation algorithm that achieves **competitive performances on CIFAR10 generation** tasks, and **lossless aesthetic performances** on the one-step distillation of the **DiT-based text-to-image diffusion model**. Besides the strong empirical performances, the theoretical foundation of SIM is also sound: we prove that the SIM loss has the same $\theta$ gradient as our proposed score-based divergence. Therefore **using an SGD-type optimization algorithm to minimize SIM loss is secretly equivalent to minimizing the intractable score-based divergence**. Besides, our proposed score-gradient theorem may bring new tools for future studies on generative models.
>
> Next, we address your concerns one by one.
>
> **Q1**. Clarify the reason for using Pseudo-Huber distance.
>
> **A1**. We appreciate your keen intuitions. Though the standard L2 distance seems a fair choice, using the Pseudo-Huber distance is better. The Phuber loss (equation 3.8) has a denominator of the form $\sqrt{\||y_t\||^2_2 + c^2}$. Here $y=s_g(x,t) - s_d(x,t)$ will be close to zero when the generator is converging. Therefore, if we set $c=0$ (the case of squared L2 distance), such a denominator is unstable and can lead to numerical issues. This means that Phuber with a small $c$ is better than the standard of L2 distance.
>
> Another advantage of Phuber loss (equation 3.8) is that it has an adaptive self-normalization effect on the loss scale. This helps the training loss to be stable with a constant scale.
>
> **Q2**. Can the authors show how SIM becomes a DMD objective if we set $d(\cdot)$ to be reversed KL?
>
> **A2**. If we consider the EDM formulation of Algorithm 2 in the Appendix, the DMD loss can be viewed as a special case of SIM loss. Let $d_{\phi}(x_t,t)$ denotes the online EDM model, and $d_q(x_t,t)$ the teacher EDM model. In Algorithm 2, the SIM loss with Phuber distance writes:
> $$L(\theta) = \frac{d\_{\phi}(x\_t,t) - d\_q(x\_t,t)}{\sqrt{\||d\_{\phi}(x\_t,t) - d\_q(x\_t,t)\||\_2^2 + c^2}} (d\_{\phi}(x\_t,t) - x\_0).$$ If we cut off the $\theta$ gradient of $x_t$, then $d_{\phi}(x_t,t)$ and $d_q(x_t,t)$ will have no parameter dependence. So the SIM loss is equivalent to DMD loss with a weighting function $\frac{1}{\sqrt{\||d_{\phi}(x_t,t) - d_q(x_t,t)\||_2^2 + c^2}}$. However, the theory behind SIM and DMD is essentially different, because SIM aims to minimize a general family of score-based divergences, instead, DMD aims to minimize the KL divergence which is notorious for mode-collapse behavior.
>
> There might be a reason why **score-based divergence is a better choice than KL**. Recall that the definition of KL divergence needs the likelihood ratio $\frac{p_g(x)}{p_d(x)}$, which will be ill-defined when $p_d$ and $p_d$ have misaligned density support. This can potentially lead to mode-collapse issues when using KL divergence as a target. However, the score-based divergence does not have such a "ratio", and therefore is safe in the case when $p_g$ and $p_d$ have misaligned support.
>
> **Q3**. The motivation to distill PixelArt-$\alpha$ diffusion model.
>
> **A3**. In Section 4.1, we evaluate the SIM on the CIFAR10 generation benchmark thoroughly and observe strong performances without using GAN losses. After that, we would like to scale SIM to challenging text-to-image generation. We choose PixelArt-$\alpha$ diffusion as a teacher model for three reasons. **(1)** We appreciate the great efforts of previous works on diffusion distillation of UNet-based t2i models. However, **distilling DiT-based diffusion models, such as PixelArt-$\alpha$ diffusion, lack explorations**. **(2)** The PixelArt-$\alpha$ model uses the T5-xxl text encoder, which is **able to understand long prompts** better than Stable Diffusion models with CLIP text encoders. We are motivated to explore high-quality one-step generator models that have such a long-prompt following ability. **(3)** We would like to explore the limits of a one-step model that is **favored by human users** in terms of Aesthetic Scores. The PixelArt-$\alpha$ is a good model that has been trained with high-quality aesthetic images.
>
> **Q4**. Missing hyperlinks and some suggestions to improve the presentation.
>
> **A4**. We feel sorry for the confusion. The hyperlinks are missing because we have separated the main pages and the appendix. We will incorporate your suggestions to improve our presentation in the revision.
>
> **Q5**. Not a major concern, but I wonder if SIM can be further improved with the SiD codebase.
>
> **A5**. We appreciate your constructive suggestion. After reading the SiD's official code, we find the official code of SiD has multiple tricks that are different from our original implementation. With some tests, we find that these tricks, such as masking out the nan samples, the weighting function, as well as the continuous-time Karras-$\rho$ noise sigma distribution help stabilize the training and improve the performance.
>
> Due to the limited time of the rebuttal period and limited compute, we conduct experiments with two settings to compare SIM and SiD behavior at different training stages: training from scratch, and resuming from a nearly converged generator; **Please check our global rebuttal and added one-page** for technical details.
>
> Our new experiment shows the faster convergence of SIM than SiD ($\alpha=1.0$) using the same implementation techniques. However, training to the end will take a very long time, and we have to acknowledge that we probably can not make a final conclusion that SIM will be better than SiD without sufficient computation. Besides, we appreciate SiD's high-quality implementations and solid contributions.
>
> **We hope our response has resolved your concerns. If you still have any concerns, please let us know.**

---

> > ### Comment · Reviewer_wPU2 · 2024-08-13
> > **Thank you**
> >
> > I thank the authors for providing detailed answers to my questions/concerns. My major concerns have been addressed so I increase my rating.

---

> > > ### Author Response · Authors · 2024-08-13
> > > **Thank you for response.**
> > >
> > > We appreciate your response and valuable suggestions. We sincerely hope that our responses have adequately addressed the concerns you raised in your review. **For any unresolved concerns or additional questions, please do not hesitate to let us know**. We would be happy to provide further clarification and address any remaining issues.

---

### Official Review · Reviewer_pChh · 2024-07-17

**Soundness:** 2
**Presentation:** 2
**Contribution:** 2
**Rating:** 5
**Confidence:** 3

**Summary:**

This paper proposes a new distribution matching loss between the one-step generator and the pre-trained diffusion model. The reverse KL divergence proposed in Diff-Instruct is generalized through the Score-divergence gradient Theorem.

**Strengths:**

1. The Score-divergence theorem is well adapted to generalize Diff-Inst.

2. The empirical results with generalized objective function show better with expanded hyperparameter space.

3. A method like CTM utilizes GAN loss (w/ true data) to boost the performance. Besides the CTM, this paper's performance seems good.

4. The distillation training cost seems cheap.

**Weaknesses:**

1. More analysis on $\alpha$ seems required.

2. The score network for the student model is still required. I think the existence of an auxiliary score network is expensive. For example, a recent paper in [1] does not utilize an auxiliary score network.

3. Adding recall metric in cifar-10 experiments if you want to claim Diff-Instruct suffers from mode-collapse.

4. Since Diff-Instruct uses reverse-KL divergence, their mode collapse is intuitive for me. Why does the objective function you propose not to suffer from mode collapse? I know why forward KL divergence can cover the mode and reverse KL can not do it. Please explain it similar sense. There may be a difference depending on the $\alpha$.

[1] Multistep Distillation of Diffusion Models via moment-matching

**Questions:**

1. Is the score-divergence theorem mentioned in the diffusion model community? It seems applicable to diffusion model training.

2. Will you release the code?

**Limitations:**

None.

---

> ### Author Rebuttal · Authors · 2024-08-07
>
> Thank you for your constructive feedback. We will address your concerns one by one. Before that, we first give a summary of the main contributions of our work.
>
> In this work, we introduce score implicit matching (SIM), a novel diffusion distillation algorithm that achieves **competitive performances on CIFAR10 generation** tasks, and **lossless aesthetic performances** on the one-step distillation of the **DiT-based text-to-image diffusion model**. Besides the strong empirical performances, the theoretical foundation of SIM is also sound: **we prove that the SIM loss shares the same $\theta$ gradient as our proposed score-based divergence**. Therefore using an SGD-type optimization algorithm to minimize SIM loss is secretly equivalent to minimizing the intractable score-based divergence. Besides, our proposed score-gradient theorem may bring **new theoretical tools for future studies on generative models**.
>
> Next, we address your concerns one by one.
>
> **Q1**. More analysis on $\alpha$.
>
> **A1**. We are not sure what you mean by $\alpha$. In the SIM algorithm (Algorithm 1), we do not have an $\alpha$ hyperparameter. We guess that you mentioned $\alpha$ to mean the SiD algorithm, which uses an empirical
> $\alpha$ hyperparameter to fuse its loss functions. SIM is different from SiD in theory. As we have shown in Sections 3.3 and 3.4, when taking the distance function to be squared L2 distance, SiD's empirical fused loss with an $\alpha=1.0$ is a special case of SIM. Besides, SIM supports a flexible choice of distance functions to define and instantiate different score-based loss functions. This distinguishes SIM from previous methods such as SiD and Diff-Instruct.
>
> **Q2**. The existence of an auxiliary score network is expensive. A recent paper in [1] does not utilize an auxiliary score network.
>
> **A2.** We appreciate your keen intuition. We acknowledge that SIM needs an additional online diffusion model, which brings more memory costs. However, in practice, we find that the **additional cost is acceptable** if we use Pytorch techniques. For example, in our text-to-image experiment, we use 4 Nvidia A100-80G GPUs for training the one-step generator. We use the Pytorch Distributed data-parallel (DDP) to support multi-GPU gradient syncing and use the BF16 numerical format for training. We can effectively train the model with a total batch size of 256 even with no gradient accumulation. If we use gradient accumulation techniques, we can enlarge the batch size to 1024 with some speed decline.
>
> However, **we highly appreciate your suggestions to refer to [1] to figure out the possibility of getting rid of the online diffusion model**. After a careful read of [1], we like the idea of moment matching to drop the online diffusion model. We will involve a discussion on SIM and [1] in the revision and leave the study that drops the online diffusion while maintaining strong one-step generation performance in our future work.
>
> **Q3**. Why does SIM not suffer from mode collapse?
>
> **A3**. We appreciate your good question! SIM is essentially different from the Diff-Instruct algorithm in the divergence that the generator aims to minimize. If we use $p_d$ to represent data distribution and $p_g$ the generator distribution, the Diff-Instruct minimizes the integral of the **KL divergence** between the generator and the teacher diffusion model with a form $$\mathcal{D}\_{KL} (p\_g, p\_d) = \mathbb{E}\_{x\sim p\_g}\log \frac{p\_g(x)}{p\_d(x)},$$ while the SIM minimizes the score-based divergence with a form $$\mathcal{D}\_{SD}(p\_g,p\_d) = \mathbb{E}\_{x\sim \pi} d(\nabla\_x \log p\_g(x) - \nabla\_x \log p\_d(x)).$$ KL divergence is notorious for mode-collapse issues because the likelihood ratio $\frac{p_g(x)}{p_d(x)}$ will be ill-defined if $p_g$ and $p_d$ have misaligned density support. However, the score-based divergence does not have such a "ratio", and therefore is safe in the case when $p_g$ and $p_d$ have misaligned support. Besides, our SIM with Phuber distance (equation 3.8)  has a self-normalization form, which helps stabilize the training loss to a constant scale which potentially addresses the mode-collapse issue.
>
> **Q4**. Is the score-divergence theorem mentioned in the diffusion model community?
>
> **A4**. As we introduced in Section 3.2, the Score-divergence gradient Theorem is proved by taking the parameter gradient to both sides of the so-called score-projection identity. The score-projection identity is well-known for proving the equivalence of denoising score matching and classical score matching in [2]. Recently, [3] also used the projection identity to prove value equality when deriving SiD loss.
>
> However, as far as we know, our score-divergence gradient theorem is the first time it appeared in the community, marking it as an important theoretical contribution in our paper. In this paper, we propose and use the score-divergence gradient theorem to prove the gradient equality of SIM loss and intractable score-based divergence instead of value equality. This distinguishes SIM from previous methods such as SiD that only prove value equality that is insufficient in theory.
>
> **Q5**. Will you release the code?
>
> **A5**. We will release the code if the paper gets published.
>
> **Q6**. Add a recall metric to show that Diff-Instruct suffers from mode-collapse.
>
> **Q6**. Thank you for your suggestion. In the rebuttal period, we compute the precision and recall of the DI model. It has a recall of 0.52 which is pretty low, indicating its mode-collapse issue.
>
> [1] Multistep Distillation of Diffusion Models via moment-matching
>
> [2] A Connection Between Score Matching and Denoising Autoencoders
>
> [3] Score identity Distillation: Exponentially Fast Distillation of Pretrained Diffusion Models for One-Step Generation
>
> **We hope our answers have resolved your concerns, and if you still have any concerns, please do let us know.**

---

### Official Review · Reviewer_G1sA · 2024-07-28

**Soundness:** 3
**Presentation:** 3
**Contribution:** 4
**Rating:** 8
**Confidence:** 4

**Summary:**

The paper ‘One-Step Diffusion Distillation through Score Implicit Matching’ introduces a novel framework Score Implicit Matching (SIM), to distill pre-trained diffusion models into single-step generator models. This approach achieves almost no performance loss compared to the teacher diffusion model while being data-free i.e. this approach does not require ground truth data.

**Key Insights**
- The paper proposes a flexible class of score-based divergences between the generator model (student) and the diffusion model (teacher). - Though such divergences cannot be computed explicitly, the gradients of these divergences can be computed via the score-gradient theorem, allowing for efficient training.
- The Pseudo-Huber distance function in SIM results in stronger robustness to hyper-parameters such as learning rate and faster convergence.
- The paper provides a detailed algorithm for implementing SIM, which involves alternating phases of learning the marginal score function and updating the generator model.

**Empirical Performance**
- On CIFAR10, SIM achieves an FID of 2.17 for unconditional generation and 1.96 for class-conditional generation.
- For text-to-image generation, a single-step generator distilled using SIM from a leading diffusion-transformer-based model attains an aesthetic score of 6.42, outperforming other one-step generators like SDXL-TURBO, SDXL-LIGHTNING, and HYPER-SDXL.

**Comparative Analysis**
- SIM is compared to other methods like Diff-Instruct (DI) and Score Identity Distillation (SiD), demonstrating better performance in terms of robustness, convergence speed, and generative quality.
- The empirical results highlight SIM’s superiority, particularly in maintaining the performance of the original multi-step models in a one-step generation framework.

**Practical Applications**
- The data-free nature and robust convergence make SIM a highly efficient method for distilling diffusion models.
- The method shows strong potential for scaling to more complex tasks and larger neural networks.

**Limitations and Future Work**
- The applicability of SIM to other generative models, such as flow-matching models, is yet to be unexplored.
- While SIM is data-free, incorporating new data might further enhance performance, which is a potential area for future research.

Overall, the paper presents a significant advancement in the field of diffusion model distillation, offering a practical and efficient method for transforming pre-trained multi-step diffusion models into one-step generators without compromising performance.

**Strengths:**

**Originality**
- The paper presents a novel approach - Score Implicit Matching (SIM) to distill pre-trained diffusion models into one-step generator models. This approach stands out by being data-free, which is a significant deviation from conventional distillation methods that often require extensive training data.
- The paper’s key technical insight, the score-gradient theorem, allows computation of gradients for score-based divergences, enabling efficient training. This is a novel contribution that enhances the feasibility of implicit minimization of these divergences.
- Pseudo-Huber Distance Function normalizes the distance vector y_t, which in turn stabilizes training loss and results in robustness to hyper-params and faster convergence.

**Quality**
- The paper provides compelling empirical evidence of SIM's effectiveness on CIFAR10 image generation. The approach also outperforms other one-step generators on aesthetic score on COCO-2017 validation set.
- The authors conduct thorough comparisons with existing methods, such as Diff-Instruct (DI) and Score Identity Distillation (SiD), showing that SIM not only outperforms these methods but also converges faster and is more robust to hyper-parameters.

**Significance**
- SIM makes a significant contribution to the practical deployment of diffusion models in real-world applications, particularly where computational efficiency is critical.
- The robustness and fast convergence of SIM suggest that it can scale to more complex tasks and larger neural networks, making it a valuable tool for future research and application in various domains, including image and video generation.
- The potential of SIM to generalize to other generative models, such as flow-matching models, opens up new avenues for research and application, further enhancing its significance in the field of generative modeling.

**Clarity**
- The paper is well-organized, with a logical flow from problem formulation to the introduction of the SIM method, followed by empirical evaluations and discussions of results.
- Key concepts, such as the score-gradient theorem and the Pseudo-Huber distance function, are explained in detail for reproducibility.
The tables and figures that present empirical results and algorithmic steps help in better understanding the performance and implementation details of SIM.


The paper presents a substantial advancement in the field of diffusion model distillation, offering a robust and efficient method for converting multi-step diffusion models into one-step generators without compromising performance. The combination of theoretical innovation with strong empirical results underscores the potential of SIM for widespread application and further research in generative modeling.
Overall, "One-Step Diffusion Distillation through Score Implicit Matching" is a well-executed study that addresses a key challenge in diffusion models and provides a promising solution with broad applicability.

**Weaknesses:**

- **Narrow Scope of Comparisons**: The empirical comparisons focus primarily on Diff-Instruct (DI) and Score Identity Distillation (SiD). While these are relevant baselines, doing a broader comparison with other state-of-the-art distillation and generative modeling techniques would provide a more comprehensive evaluation of SIM's relative performance and applicability.
- **Dataset Limitation**: The experiments are mainly conducted on the CIFAR-10, SAM-LLaVA-Caption10M and COCO-2017 validation dataset. Including results on more diverse datasets, such as higher resolution images, different image domains (e.g., medical imaging, satellite imagery), and more complex text-to-image datasets, would strengthen the claim of SIM's broad applicability.
- **Addressing Failure Modes**: The paper does not thoroughly discuss the potential failure modes or limitations of the SIM method. Identifying scenarios where SIM might underperform, and providing insights or hypotheses on why these failures might occur, would be valuable for understanding the boundaries of the method’s applicability and for guiding future improvements.
- **Exploration of Data Incorporation**: While the data-free nature of SIM is highlighted as a strength, the potential benefits of incorporating new data during distillation are mentioned but not explored. Providing preliminary experiments or a more detailed discussion on how incorporating data might enhance the quality of the distilled models would be a constructive addition.

The paper presents significant contributions to the field of diffusion models. Addressing the above weaknesses would further enhance its impact and applicability.

**Questions:**

Some of these are already mentioned under weakness.
- Evaluating the performance of SIM on different image domains would broaden the applicability of this approach
- Discussing scenarios where SIM would not perform well would help in understanding the boundaries and limitations for this approach and guide future improvements.

Additionally, it would be helpful if the authors can provide mode implementation details such as hyper-parameter tuning strategies and potential pitfalls during the training process.

Misc
- In the appendix, it is mentioned that a human preference study was conducted and the authors collected 30 responses in total. It would be valuable to know the results from the human preference study.
- Grammar/Typos
   - Line 130 - ‘... if we choose the sampling distribution to be the diffused ..’
   - Line 220 appears to be grammatically incorrect - ‘It is on par with the CTM and the SiD’s official implementation has yet to be released’

**Limitations:**

The authors have acknowledged limitations of the proposed approach.

- The applicability of SIM has not been explored in the space of other generative models such as flow-matching models.
- Although the approach is data-free, the potential benefit of incorporating data is yet to be explored.

I do not see potential negative societal impacts of this work.

---

> ### Author Rebuttal · Authors · 2024-08-07
>
> We are glad that you like our work. We appreciate your valuable suggestions, and we will take them in the revision. In the following paragraphs, we will address your concerns one by one.
>
> **Q1**. Doing a broader comparison with other distillation and generative modeling techniques would provide a more comprehensive evaluation of SIM's relative performance and applicability.
>
> **A1**. In this paper, we have compared SIM with Diff-Instruct and SiD, showing its soundness on both the theoretical and empirical side. Another leading diffusion distillation method might be the consistency trajectory model (CTM) [1], which uses self-consistency to distill the model. Though CTM has shown strong performances, its performances rely on using additional LPIPS losses and GAN losses, which need ground truth data and are also tricky. Besides, the CTM model requires running multiple steps of reversed diffusion sampling, which is more computationally inefficient than SIM without running multiple-step sampling with teacher diffusion. We will compare with other leading distillation methods in the revision.
>
> **Q2**. Including results on more diverse datasets, such as medical imaging, and satellite imagery would strengthen the claim of SIM's broad applicability.
>
> **A2**. We highly appreciate your suggestions to use SIM on broader applications. In the rebuttal period, we conduct two more experiments: (1) text-to-3D generation using text-to-2D diffusion; and (2) improving the GAN generator using diffusion models; **Please check our Global rebuttal cell for details.** On both new applications, SIM has shown strong performances, showing its broad applicability.
>
> **Q3**. Providing preliminary experiments or a more detailed discussion on how incorporating data might enhance the quality of the distilled models would be a constructive addition.
>
> **A3**. We appreciate your keen intuition. One advantage of SIM is the image-data-free property. However, we agree that incorporating training data with SIM will potentially strengthen the SIM in real-world applications. A possible way to incorporate data is to use additional GAN discriminator and GAN losses together with the SIM loss. This can inject knowledge from new data into the generator model. Another direction that is worth exploring is to extend the one-step generator to multi-step generative models using variants of SIM. This potentially will combine techniques with consistency-based models. However, both the GAN loss and the multi-step generalization are more like a combination of different helpful techniques in generative modeling. In this submission, our goal is to thoroughly study the theory and practical performance of SIM. We will leave the study that incorporates data with SIM in the future.
>
> **Q4**. Discussing scenarios where SIM would not perform well would help in understanding the boundaries and limitations of this approach and guide future improvements.
>
> **A4**. We appreciate your constructive suggestion. In our text-to-image experiment, we find our SIM-DiT-600M sometimes fails to generate high-quality tiny human faces, hands, as well as legs. However, we find that the teacher PixelArt-$\alpha$ model also has such issues. We believe that a stronger teacher diffusion model will possibly address such failure cases. Please check **Figure 2** in our **added one-page** in global rebuttal for some failure cases.
>
> **Q5**. The human preference study.
>
> **A5**. We feel sorry for the confusion, we put the human preference comparison of SIM-DiT-600M and PixelArt-$\alpha$ diffusion model in **Table 3** in our submission. The result shows that the teacher PixelArt-$\alpha$ model slightly outperforms the SIM one-step model with a winning rate of 54.88\%. We will polish the presentation in the revision.
>
> **Q6**. Implementation details.
>
> **A6**. We will release the code if the paper gets published. In this response, we add some more implementation details for the one-step text-to-image generator model. When distilling the generator, we first reformulate the PixelArt-$\alpha$ diffusion with the so-called "data-prediction" formulation as proposed in the EDM paper [2]. We then initialize our generator using the "data-prediction" model with a fixed noise sigma of 2.5, following the same setting of Diff-Instruct and SiD. When distillation, we use a fixed classifier-free guidance scale of 4.5 for teacher diffusion. For the auxiliary diffusion model, we do not use the CFG. This setting is inspired by the DMD paper [3]. We will add more implementation details in the Appendix in the revision.
>
> **Q7**. Some typos.
>
> **A7**. We will address the typos and improve our writing in the revision.
>
> [1] Consistency Trajectory Models: Learning Probability Flow ODE Trajectory of Diffusion
>
> [2] Elucidating the Design Space of Diffusion-Based Generative Models
>
> [3] One-step Diffusion with Distribution Matching Distillation
>
> **We hope our response has resolved your concerns. If you still have any concerns, please let us know.**

---

> > ### Comment · Reviewer_G1sA · 2024-08-13
> > **Acknowledging the Rebuttal**
> >
> > Thank you for the response. I acknowledge the rebuttal. I still maintain my score as 8 -- Strong Accept.

---

### Author Rebuttal · Authors · 2024-08-07

We appreciate all reviewers for your valuable feedback. In this cell, we address some common concerns.

As the **Reviewer G1sA** and **Reviewer Jq2C** wish, we run two additional experiments to demonstrate the wide applicability of SIM: (1) text-to-3D generation using text-to-2D diffusion; and (2) improving the GAN generator using diffusion models; On both new applications, SIM has shown strong performances, demonstrating its broad applicability.

**(1) text-to-3D generation.** We follow the setting of ProlificDreamer which minimizes the KL divergence in order for 3D generation. We use the open-sourced ThreeStudio Pytorch implementation of ProlificDreamer.

Result: **Figure 3** of our **added one-page** in the global response shows a qualitative comparison of SIM and ProlificDreamer (VSD). We find that when using SIM for text-to-3D generations leads to decent results, which is visually comparable with ProlificDreamer. This result shows that SIM has the potential to be used for text-to-3D generation. However, since our paper aims to focus on one-step image generation, and the computation for a thorough quantitative evaluation for 3D generation is expensive, we only give the qualitative visualization results in the rebuttal period. We leave more study and tuning on using SIM for 3D generation in our future work.

**(2) improving StyleGAN2 generator.** In this experiment, we use SIM to improve a pre-trained stylegan2 generator with pre-trained EDM diffusion models, following the same settings as the Diff-Instruct [2] paper's Section 4.2. **Table 1** and **Table 2** in our **added one-page** in the global response show the quantitative results of improving the generator experiment. As a result, we surprisingly find that SIM is able to improve the pre-trained StyleGAN2 generator better than Diff-Instruct can. This shows that SIM has a high potential to improve existing GAN generators, such as StyleGAN-T and GigaGAN. We will leave the scaling-up work to improve text-to-image GAN models in future work.


[1] ProlificDreamer: High-Fidelity and Diverse Text-to-3D Generation with Variational Score Distillation

[2] Diff-Instruct: A Universal Approach for Transferring Knowledge From Pre-trained Diffusion Models


As **Reviewer WiWu** and **Reviewer wPU2** wish, considering the limited time of the rebuttal period, we run two new experiments to compare SIM and SiD on **CIFAR10 unconditional generation** task behavior at different training stages: (3) training from scratch, and (4) resuming from a nearly converged generator;

**(3) compare SIM and SiD from scratch.** We compare SIM and SiD ($\alpha=1.0$) with SiD's official code and hyper-parameters to train a one-step generator from scratch. Both training runs are on 8 Nvidia-A100-80G GPUs with a total batch size of 128. We use the default settings as SiD, with the only difference being the use of loss functions. We log the FID value along the training process as a performance metric. **Figure 1** in our **added one-page** shows the FID Comparison of SIM and SiD for training from scratch.

**Result**: both methods can converge steadily. The SIM converges faster than SiD, with a lower FID value at the same number of training iterations.

**Findings**: We find that the weighting function that SiD uses is important for stable training. We find that using this weighting function for SIM can improve SIM's performance compared with our original implementation, which makes us appreciate the technical contribution of SiD authors for their high-quality implementations. We also find that training a generator to a very low FID is slow for both SIM and SiD. Due to the limited time of rebuttal period, this motivates us for our second experiment.

**(4) compare SIM and SiD by resuming from a nearly-converged generator.** In this experiment, we compare the convergence speed with SIM and SiD in the case of resuming from a nearly-converged one-step generator. Because a full training trial is very slow (it may last for up to several weeks with 16 GPUs). Therefore training until the end is impossible for the short rebuttal period. Therefore, we resume the generator from a checkpoint that we previously trained with our SiD implementation which has an FID of 2.8. We aim to inspect the training behavior of SIM and SiD near convergence with this experiment. For this experiment, we use 4 Nvidia-H800 GPUs with a batch size of 128 for both SiD and SIM. All hyperparameters of SIM are set the same as SiD's default choices except for a learning rate of 1e-6.

**Result**: **Figure 1** in our **added one-page** shows the FID Comparison of SIM and SiD. Both methods can converge slowly near convergence. The SIM converges faster than SiD, with a lower FID value with the same compute. With a couple of days of training, the SIM is able to reach a minimum FID of 2.06 from 2.8, while the SiD's minimum FID is larger than 2.15.

Next, we will address each reviewer's concerns in individual rebuttal cells.

---

### Decision · Program_Chairs · 2024-09-25

**Decision:**

Accept (poster)

**Comment:**

Reviewers agreed that this work presents a robust and efficient method for distillation into one-step generators. While there were some concerns on experimental differences with prior work, the experimental results on large-scale text-to-image models are compelling and demonstrate effective one-step distillation.